# Tau uptake by human neurons depends on receptor LRP1 and kinase LRRK2

Lewis D Evans [ID][1,2,3], Alessio Strano[1,4], Eleanor Tuck[1,2,3], Ashley Campbell[1,2,3], James Smith [ID][1,5], Christy Hung [ID][1,6], Tiana S Behr [ID][7], Bernardino Ghetti[8], Benjamin Ryskeldi-Falcon[7], Emre Karakoc[2], Francesco Iorio [ID][2,9], Alastair Reith [ID][10], Andrew R Bassett [ID][2,3] & Frederick J Livesey [ID][1,5 ✉]

## Abstract

Extracellular release and uptake of pathogenic forms of the microtubule-associated protein tau contribute to the pathogenesis of several neurodegenerative diseases, including Alzheimer's disease. Defining the cellular mechanisms and pathways for tau entry to human neurons is essential to understanding tauopathy pathogenesis and enabling the rational design of disease-modifying therapeutics. Here, whole-genome, loss-of-function CRISPR screens in human iPSC-derived excitatory neurons, the major neuronal cell type affected in these diseases, provide insights into the different cellular pathways for uptake of extracellular monomeric and fibrillar tau. Monomeric and fibrillar tau are both taken up by human neurons by receptor-mediated endocytosis, but involve different routes of entry at the neuronal surface: the low-density lipoprotein LRP1 is the primary receptor for monomeric tau, but contributes less to fibrillar tau entry. Similarly, endocytosis of monomeric tau is dependent on the familial Parkinson's disease gene LRRK2, but not required for endocytosis of fibrillar tau. These findings implicate LRP1 and LRRK2 in the pathogenesis of tauopathies and Parkinson's disease, and identify LRRK2 as a potential therapeutic target for altering progression of these diseases.

**Keywords** Alzheimer's Disease; Parkinson's Disease; CRISPR; Functional Genomics; Human iPSCs
**Subject Categories** Membranes & Trafficking; Molecular Biology of Disease; Neuroscience

## Introduction

Neurodegenerative diseases that involve the microtubule-associated protein tau (tau, *MAPT*), such as Alzheimer's disease, corticobasal degeneration and progressive supranuclear palsy, are referred to collectively as tauopathies (Lee et al, 2001). Characterised by the formation of intracellular aggregates of tau protein, these diseases also display stereotyped spatiotemporal progression of pathology through the central nervous system, including the cerebral cortex (Goedert et al, 2017; Braak and Braak, 1991). The spatial progression of tau aggregation mirrors the connectivity of the central nervous system (Vogel et al, 2020), and the involvement of different brain regions at different stages of disease is reflected in the development of clinical symptoms. A current working model for tauopathy progression is that pathogenic forms of tau protein are released from diseased neurons containing tau protein aggregates. These are either taken up by synaptically-connected neurons or processed and released by microglia before subsequent uptake by neurons (Asai et al, 2015). In both cases, pathogenic forms of tau are hypothesised to be transferred to surrounding neurons, seeding further tau aggregation and neuronal dysfunction (Braak and Braak, 1991).

There is considerable interest in identifying the forms of tau and associated molecules that mediate disease progression and the biological mechanisms involved in potential intercellular tau transfer, including cell surface receptors. In addition to understanding disease pathogenesis, such insights are essential to the rational design of therapies to slow disease progression (Vogel et al, 2020). It is currently not known which forms of tau mediate the proposed pathogenic spreading of tauopathy between neurons (Kaniyappan et al, 2020), although oligomers and fibrillar aggregates of either full-length tau or fragments that include the microtubule-binding region are considered strong candidates (Walsh and Selkoe, 2016). While the identity of the pathogenic tau species responsible for disease propagation remains unknown, progress has been made in understanding the mechanisms of cellular uptake of tau (Goedert et al, 2017). Cell surface heparan sulfate proteoglycans have been found to be necessary for uptake of tau fibrillar aggregates (Holmes et al, 2013), and we and others have found that human neurons efficiently take up both monomeric and fibrillar tau by overlapping but distinct mechanisms, consistent with dynamin-dependent receptor-mediated endocytosis (Evans et al, 2018; Rauch et al, 2018). More recently, the low-density

[1]UCL Great Ormond Street Institute of Child Health, Department of Developmental Biology and Cancer, Zayed Centre for Research into Rare Disease in Children, London WC1N 1DZ, UK. [2]Open Targets, Wellcome Genome Campus, Cambridge CB10 1SA, UK. [3]Wellcome Sanger Institute, Wellcome Genome Campus, Cambridge CB10 1SA, UK. [4]Department of Biosystems Science and Engineering, ETH Zurich, Basel, Switzerland. [5]Talisman Therapeutics, Babraham Research Campus, Cambridge CB22 3AT, UK. [6]Department of Neuroscience, City University of Hong Kong, Hong Kong, Hong Kong SAR. [7]MRC Laboratory of Molecular Biology, Cambridge CB2 0QH, UK. [8]Department of Pathology and Laboratory Medicine, Indiana University School of Medicine, Indianapolis, IN, USA. [9]Human Technopole, 20157 Milano, Italy. [10]Novel Human Genetics Research Unit, GlaxoSmithKline Pharmaceuticals R&D, Stevenage, UK. ✉E-mail: rick@talisman-therapeutics.com

lipoprotein receptor LRP1 has been identified as a major neuronal receptor for monomeric and oligomeric tau uptake via receptor-mediated endocytosis (Rauch et al, 2020).

To define the cell biology of neuronal uptake of extracellular tau, we used whole-genome CRISPR loss-of-function screens in human iPSC-derived cortical excitatory neurons to identify genes and pathways required for neuronal uptake of tau protein. Given the accumulating evidence that different forms of monomeric, oligomeric and fibrillar tau may contribute to disease pathogenesis (Clavaguera et al, 2009; Frost et al, 2009; Sato et al, 2018), we carried out whole-genome CRISPR knockout screens (Tzelepis et al, 2016) for genes required for neuronal uptake of either full-length monomer or fibrillar aggregates of tau (Evans et al, 2018). Previous work has indicated that tau monomers and oligomers are both endocytosed via the LRP1 receptor (Rauch et al, 2020), thus, for screening purposes, we used monomeric full-length tau to represent both monomeric and oligomeric tau. We also carried out screens for uptake of fibrillar, aggregated tau, which we and others have previously shown enters human neurons via a pathway distinct from that of monomeric tau (Evans et al, 2018; McEwan et al, 2017). These screens were performed using human iPSC-derived cerebral cortex excitatory neurons (Shi et al, 2012b), the primary cell type affected by tau aggregation into neurofibrillary tangles in vivo (Lee et al, 2001). Intraneuronal levels of internalised, fluorescently-labelled tau were assessed acutely, limiting the relative importance of degradation of tau protein. As reported in detail below, genome-wide loss-of-function genetic screens identified key elements of the different pathways for neuronal entry used by monomeric and fibrillar tau, providing tractable therapeutic targets for disease intervention.

# Results

## FACS-based CRISPR knockout screens in human neurons to identify genes and pathways required for the uptake of extracellular tau

We have previously found that fibrillar and monomeric tau are taken up by human neurons by overlapping, but distinct pathways (Evans et al, 2018). CRISPR knockout screens of human neuronal uptake of extracellular tau were designed using the approach outlined (Fig. 1A). Cerebral cortex progenitor cells were generated from human iPSCs constitutively expressing Cas9 from the AAVS1 locus, using our previously described methods (Shi et al, 2012a), and differentiated to cortical excitatory neurons (Figs. 1B and EV1). Using a lentiviral reporter of Cas9 activity, we found that gene editing of the KOLF2-C1 Cas9 neurons increased over time to >35% efficiency (Fig. 1C,D).

To identify and compare the cellular pathways by which monomeric and fibrillar P301S tau enter neurons, FACS-based assays were optimised to measure uptake of both forms of tau by human iPSC-derived cortical excitatory neurons. To focus the screens on genes required specifically for tau uptake, and not general mechanisms of receptor-mediated endocytosis, assays were designed to measure uptake of transferrin and tau by the same neurons (Fig. 1). Transferrin endocytosis was confirmed not to affect tau uptake at a range of concentrations (Fig. 1E,F), and thus not interfere with the screen readout.

For the screens, a lentivirus library composed of 100,090 gRNAs, targeting 18,025 genes (Tzelepis et al, 2016), was introduced by transducing cortical progenitor cells with lentivirus at an MOI of 0.3, and progenitor cells subsequently differentiated to cortical excitatory neurons for 30 days. To identify lentivirus-transduced, guide RNA-expressing neurons that failed to take up tau but remained competent to endocytose transferrin, neurons were exposed to extracellular tau (labelled with Dylight 488) and transferrin (conjugated with Alexa 633) for either 4 or 5 h, before being dissociated to single cells and fixed. Lentivirus-transduced, gRNA-expressing neurons were gated, and the populations of neurons with high transferrin protein uptake that also had either high or low tau uptake were collected (Figs. 1G and EV1; Table EV1) and gRNA abundance in each population was measured using high-throughput sequencing.

## CRISPR screens identify genes and pathways required for human neuronal uptake of monomeric and fibrillar tau

Replicate screens for monomeric and fibrillar tau uptake were analysed using the MAGeCK algorithm (Li et al, 2014) to identify genes whose loss of function results in reduced neuronal tau uptake (Figs. 2A,B and EV2). Applying a significance cutoff of $p < 0.01$ identified 214 genes required for monomeric tau uptake and 228 genes required for fibrillar tau uptake. The low-density lipoprotein receptor LRP1, which has been shown to act as a tau receptor (Rauch et al, 2020), was the second-highest-ranked gene identified in the monomeric tau uptake screens, supporting the validity of the loss of function screen design.

For monomeric tau uptake, the most highly ranked gene was *LRRK2*, a large multifunctional protein that regulates diverse intracellular vesicle trafficking processes, mutations in which are a cause of autosomal dominant Parkinson's disease (PD) (Taylor and Alessi, 2020). In addition to *LRRK2*, there was a notable number of genes required for monomeric tau uptake encoding regulators of endocytosis, including AP2 subunits (*AP2M1*, *AP2S1*), dynamin-2 (*DNM2*), clathrin heavy chain (*CLTC*), *RAB7A*, *HGS* and PI3-kinase subunits C3 and R4. Several genes involved in endosomal sorting, such as *SNX16*, and specifically recycling of the LRP1 receptor, were also required for monomeric tau uptake, including *SNX17* (McNally et al, 2017) and the LDL receptor chaperone *MESD/MESDC2* (Hoshi et al, 2013). In addition, genes involved in lysosome and autophagosome biogenesis and acidification were prominent, including the vATPase subunit *ATP6V1D*, the vATPase assembly protein *VMA21* (Hill and Stevens, 1994), and the autophagy regulating proteins UVRAG and ATG4A (Liang et al, 2007).

In addition to individual genes, gene ontology and pathway analyses of the set of genes required for monomeric tau uptake provide a useful overview of the cellular mechanisms involved. Gene ontology and functional analysis of the set of genes above the empirical significance threshold of $p < 0.01$ found significant enrichment in several categories related to receptor-mediated endocytosis, including lipoprotein particle receptor binding, the endosome, clathrin-coated pit and the autolysosome (Fig. 2B).

Using the same significance cutoff of $p < 0.01$ as for analysis of the monomeric tau screen, identified 228 genes required for fibrillar tau uptake (Fig. 2C). *LRP1* did not reach that threshold ($p = 0.024$), suggesting that it is not a major receptor for fibrillar tau. As for

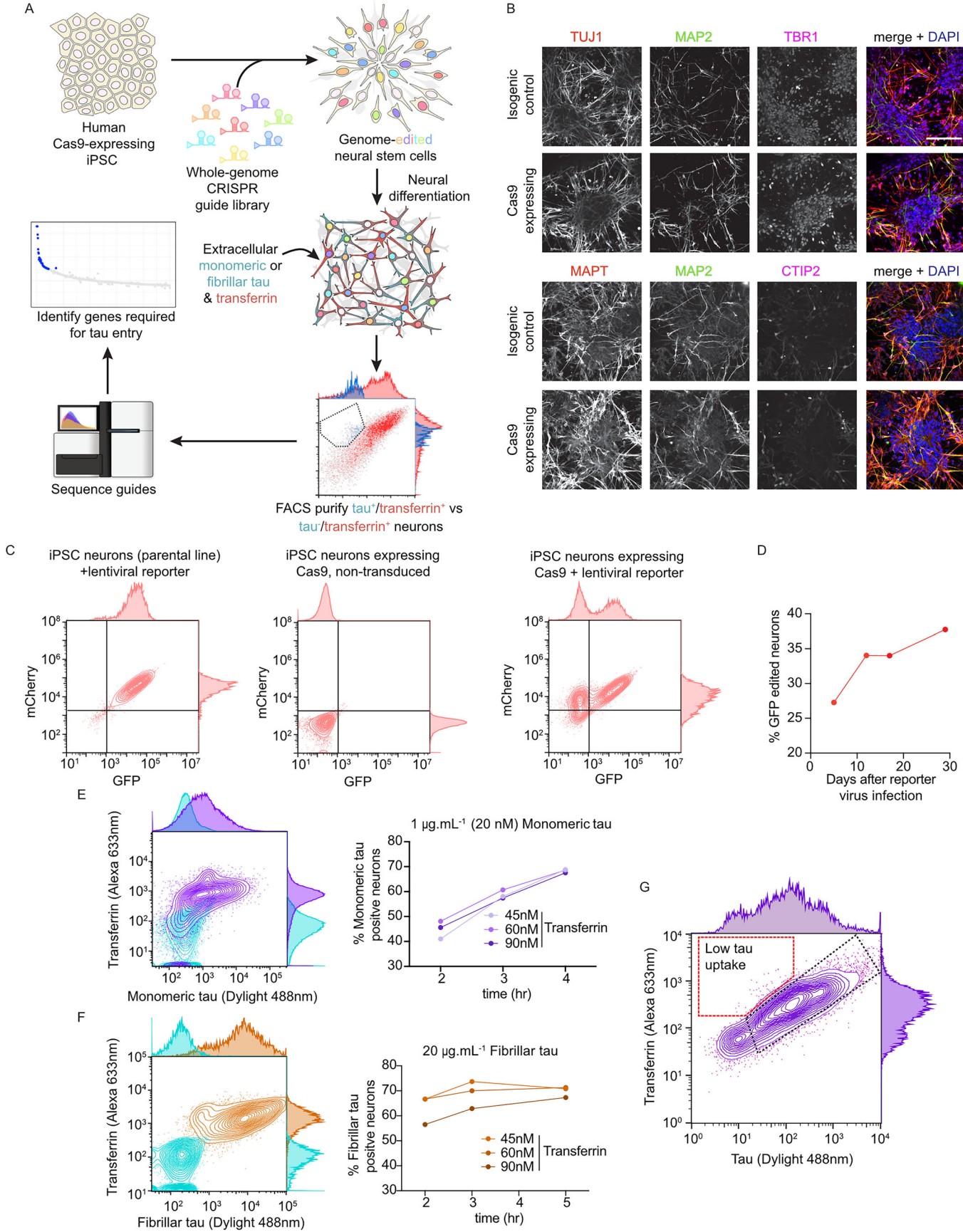

**Figure 1. FACS-based CRISPR knockout screens in human neurons to identify genes and pathways required for uptake of extracellular tau.**

(A) Design of whole-genome CRISPR knockout screens for uptake of tau by human iPSC-derived neurons. (B) Representative confocal microscopy images confirming the cortical, excitatory identity of differentiated human (KOLF2-C1) iPSC-derived neurons, constitutively expressing Cas9 protein and the isogenic control. Upper panels, neurons immunolabelled for TUJ1 (neuron-specific β3-tubulin; red), MAP2 (neuronal dendrites; green), TBR1 (deep layer neurons; magenta) and DAPI (nuclear DNA; blue). Lower panels, neurons immunolabelled for MAPT (tau protein; red), MAP2 (green), CTIP2 (deep layer neurons; magenta) and DAPI (blue). Scale bar, 50 μm. (C) iPSC-derived cortical neural progenitor cells and neurons retain Cas9 activity after differentiation. Scatter plot of FACS measuring Cas9 activity using a lentiviral Cas9-activity reporter, expressing mCherry and GFP. Left panel, iPSC-derived neurons (parental control line) transduced with lentiviral reporter but not expressing Cas9, middle panel neurons expressing Cas9 without lentiviral reporter and right panel neurons expressing Cas9 with lentiviral reporter. (D) FACS analysis of the percentage of GFP gene-edited neurons against days after reporter infection, measured using the reporter system described in (C). Data shown from one representative experiment. (E, F) Transferrin does not compete with or alter extracellular tau uptake by human excitatory neurons. Human iPSC-derived neurons were incubated with monomeric (E; purple) or fibrillar (F; orange) tau protein (Dylight 488 labelled) and transferrin (Alexa 633 conjugate; concentrations indicated) and analysed by flow cytometry. Neurons without tau or transferrin incubation (light blue) were used to establish the threshold level for detection of tau-Dylight fluorescence. Percentage of monomeric or fibrillar tau-positive neurons at indicated transferrin concentrations plotted over time after protein incubation initiation. Data shown from one representative experiment. (G) Representative scatter plot of FACS analysis of population of neurons from typical screen outcome, highlighting neurons expressing gRNAs (gated on BFP signal) that are tau-negative but transferrin-positive (dashed red line) and tau-positive and transferrin-positive (dashed black line). See also Fig. EV1.

monomeric tau, genes required for neuronal uptake of fibrillar tau include regulators of endocytosis, but in this case, *EEA1* and *HGS*, and the PI3-kinase subunits C3 and R4. Similarly, genes encoding vATPase subunits were required for fibrillar tau uptake (*ATP6V0E2*, *ATP6V0D1* and *ATP6V0A1*), as were autophagy genes such as *GABARAPL1* and *UVRAG* (Fig. 2C). Several genes encoding proteins involved in intracellular vesicular trafficking were required for fibrillar tau uptake, including the AP-1 and AP-3 complexes (*AP1G1* and *AP3S1*), and the COG complex that regulates retrograde trafficking within the Golgi (D'Souza et al, 2020) (four of the eight COG subunits: 1, 4, 7, and 8). Gene ontology and functional analysis identified categories enriched in the set of genes required for fibrillar tau uptake, including endosomal transport, the COG complex, Golgi vesicle transport, vacuolar acidification, the autophagosome and the CCC-WASH complex (Fig. 2D).

A subset of genes identified as required for uptake of monomeric tau was subsequently validated by targeted, lentivirus-mediated CRISPR knockout in iPSC-derived neural progenitor cells, followed by differentiation to excitatory neurons (Figs. 2E,F and EV2H). The effect of each CRISPR knockout on uptake of extracellular fluorescently-labelled tau and transferrin was measured by FACS, using the primary metric of the fraction of neurons with reduced tau uptake (tau-low/transferrin+ neurons) relative to the population of neurons that took up both tau and transferrin (Fig. 2E, example shown of AP2M1). Validated genes using this assay included the LRP1 receptor, both AP2 adaptor subunits AP2M1 and AP2S1, dynamin-2 (DNM2) and two PI3-kinase components (PIK3C3 and PIK3R4).

LRP1 has recently been identified as a major tau receptor for both monomeric and oligomeric tau (Rauch et al, 2020), and was one of the two most significant genes identified as required for monomeric tau entry in our screens. We confirmed that LRP1 is required for monomeric tau uptake by CRISPR knockout of LRP1 in iPSC-derived neural progenitor cells, using the FACS-assay for uptake of extracellular tau and transferrin (Fig. 2F). However, LRP1 did not meet the threshold for inclusion as a gene required for fibrillar tau entry (ranked 522 of 18,019 genes; $p = 0.024$), consistent with previous reports (Rauch et al, 2020).

To test the dependency of monomeric and fibrillar tau entry on interaction with LRP1, we used two different approaches to inhibiting tau-LRP1 interactions, combined with live-imaging of neuronal uptake of pHrodo-labelled tau. Blockade of all LDL

receptors by addition of the pan-LDL receptor chaperone RAP (Herz et al, 1991) reduced uptake of monomeric tau by ~63%, with a minor effect on fibrillar tau (Figs. 2G–I and EV2). As an alternative approach to inhibiting tau interactions with cell surface LRP1, we added soluble recombinant domain IV of LRP1, the region defined as binding tau (Rauch et al, 2020), together with pHrodo-tau, and subsequently imaged tau uptake (Figs. 2G–I and EV2). As with RAP, the addition of the LRP1 fragment reduced uptake of monomeric tau by ~68% (Fig. 2G–I), confirming LRP1 as a major receptor for neuronal uptake of monomeric but not fibrillar tau. Finally, we tested whether synthetic, recombinant fibrillar tau and in vivo AD human brain tau fibrils differ in their use of LRP1 for neuronal entry. To do so, we analysed neuronal uptake of pHrodo-labelled fibrillar tau immunopurified from post-mortem human temporal cortex of individuals who had clinically confirmed sporadic Alzheimer's disease (Falcon et al, 2018). As with synthetic tau fibrils, inhibiting LRP1 interactions with either RAP or domain IV of LRP1 did not significantly reduce the uptake of post-mortem tau aggregates/ filaments, providing further evidence that tau fibrils do not use LRP1 as a major receptor for neuronal endocytosis.

## Monomeric and fibrillar tau enter neurons via distinct endocytosis pathways

To assess the degree of similarity between the mechanisms of neuronal uptake of both forms of tau, we analysed overlaps between the results of both screens at the levels of genes and protein complexes (Figs. 3A,B and EV3). To do so, we compared the gene sets identified above from the monomeric and fibrillar tau entry screens, using the $p < 0.01$ cutoff. There was a small but significant overlap in a subset of individual genes identified by the screens for monomeric and fibrillar tau uptake (hypergeometric test $p < 0.001$; Fig. 3A), providing evidence for some shared aspects of neuronal uptake of each form of tau. Genes common to both screens included the COG complex member *COG4*, the three class III PI3K complex members discussed above (*PIK3C3, PIK3R4* and *UVRAG*), and the tyrosine kinase *HGS*, which is a known downstream effector of PI3 kinase, an ESCRT component and regulator of endosome trafficking (Raiborg and Stenmark, 2009).

In addition to individual genes, genes that contribute to four protein complexes (Fig. 3) were significantly enriched (FDR <0.05) among combined hits of the monomeric and fibrillar tau uptake screens (Fig. 3B). These four were the conserved oligomeric golgi

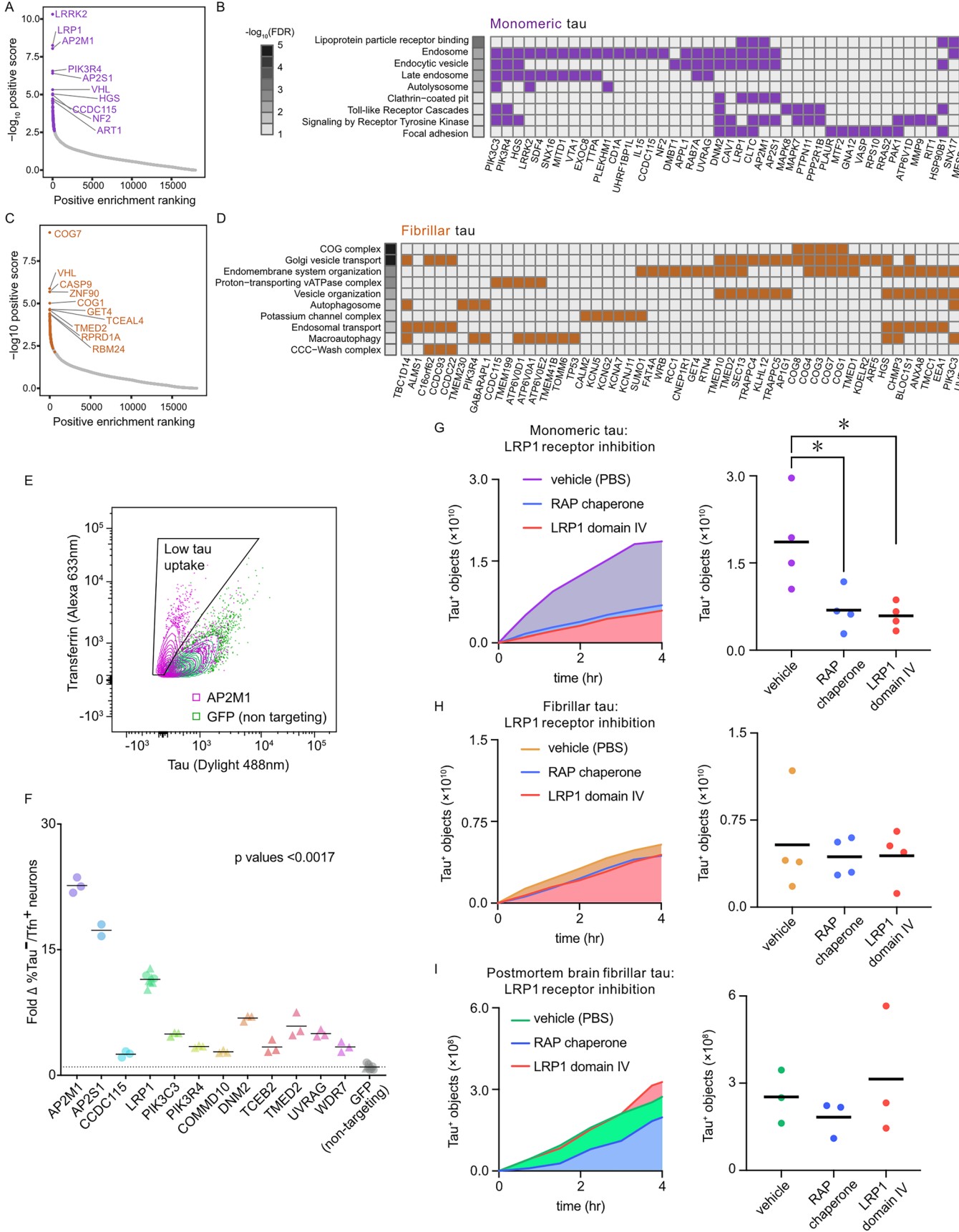

**Figure 2. CRISPR loss of function screens identify genes and pathways required for human neuronal uptake of extracellular monomeric and fibrillar tau.**

Genes required for uptake of tau into human iPSC-derived excitatory neurons identified by comparing FACS collected populations of CRISPR edited neurons, positive for labelled transferrin, but not tau protein, against neurons positive for both labelled proteins. Whole-genome enrichment score (−log10 positive score) is plotted against the positive enrichment ranking from the knockout screens for tau uptake derived using the MAGeCK algorithm. Applying a p value cut-off of 0.01, 214 genes were identified as required for (A) monomeric tau uptake (purple points; two biological replicates) and 228 genes were identified as required for (C) fibrillar tau uptake (orange points; three biological replicates). The ten highest-scoring genes are labelled. Heat maps showing a representative selection of significantly enriched terms annotating the genes required for (B) monomeric and (D) fibrillar tau uptake. Grayscale indicates significance level (−log10 FDR). Rows are sorted in order of significance; see Dataset EV1 for ranked genes required for tau uptake (p < 0.01). Each heatmap shows which of the enriched genes is annotated with each term (Dataset EV2). Details of p values for each enriched term are provided in Dataset EV2. (E) Scatter plot of FACS analysis of monomeric tau and transferrin uptake by iPSC neurons individually targeted for CRISPR knockout of AP2M1 or non-targeting control (GFP). (F) Validation of genes identified in primary screens as required for monomeric tau uptake. The percentage of low tau neurons (tau-/transferrin+) is expressed as fold change from the non-targeting control (GFP). Three technical replicates were performed, across two independent experiments, represented by circles and triangles. CRISPR knockout of LRP1 and a non-targeting control (GFP) was included in both experiments (Dataset EV5). Statistical significance was determined using one-way ANOVA, Dunnett's test for multiple comparisons (all gene perturbations presented p < 0.0001, except CCDC115 p = 0.0017). (G–I) Entry of monomeric tau into neurons is mediated by LRP1, inhibited by the addition of extracellular RAP chaperone or LRP1 domain IV peptide. The number of tau-positive objects detected over 4 h by time-lapse imaging of iPSC-derived human excitatory neurons was used to assess neuronal uptake of monomeric (A), fibrillar (B) or post-mortem AD brain (C) tau-pHrodo. Object measurements are displayed over time. Twenty-four independent measurements were taken from at least three technical replicates at 45-min intervals. Statistical significance was determined using one-way ANOVA, Dunnett's test for multiple comparisons (monomeric tau uptake with RAP chaperone *p = 0.0228; LRP1 domain IV *p = 0.0152). See also Fig. EV2.

(COG) complex, involved in vesicular trafficking within the Golgi, the vesicular ATPase that acidifies the lysosome and autophagosome (Fig. 3C), the PIK3 complex discussed above, and the CCC complex, which regulates endosomal recycling, including recycling of receptors (Bartuzi et al, 2016).

The set of 431 proteins coded by genes significantly enriched in either monomeric or aggregated uptake tau screens were analysed for known direct protein–protein interactions (Fig. EV3). That analysis generated a network of connected genes that highlights pathways responsible for tau uptake and facilitates the identification of densely connected proteins. In agreement with the gene ontology and functional analysis, there is a large subnetwork of endosomal and endocytosis proteins, which includes clathrin heavy chain (CLTC), dynamin-2 (DNM2), EEA1 and AP2 subunits. This subnetwork is part of a larger set of interactions that includes RAB7A, LRRK2, PIK3C3 and PIK3R4, all of which have roles in regulating endocytosis and endosome trafficking (Taylor and Alessi, 2020; Bilanges et al, 2019). There are other notable protein interaction subnetworks that regulate vesicle trafficking, endosome sorting, Golgi vesicle trafficking and vacuolar ATPase function. Given that acidification of late endosomes and lysosomes is a key requirement for receptor recycling, disruption of that process is likely to have a considerable impact on surface receptor number and composition (Cullen and Steinberg, 2018). Notably, LRRK2 is the most densely connected protein within the protein–protein interaction network. LRRK2 interacts directly with two of the five most connected proteins, RAB7A and CLTC. Overall, the number and density of protein–protein interactions in the tau uptake network that mediate endocytosis, endolysosome function and vesicular traffic underscore the importance of receptor-mediated endocytosis for neuronal uptake of extracellular tau.

Predicted tau uptake pathways can be delineated by assigning cellular locations to the proteins coded by genes identified in our tau uptake screens using the COMPARTMENTS dataset (Fig. EV3). In agreement with the ontology and pathway analyses, monomeric and fibrillar tau show differences in their modes of uptake. Initial interactions between either monomeric or fibrillar tau and the cell surface or plasma membrane require distinct proteins. Once tau enters neurons, it is processed by the endolysosomal system, as several proteins required for tau uptake are localised to endosomal,

late endosomal and lysosomal compartments. Proteins in the endolysosomal system required for monomeric or fibrillar tau uptake share similarities, including class III phosphatidylinositol 3-kinase (PIK3s) family proteins, but also some notable differences that may reflect distinct processing pathways for forms of tau. Fibrillar tau uptake shows a greater dependency on genes associated with the Golgi apparatus and ER-Golgi trafficking compared with monomeric tau uptake, indicating differences in post-translational modifications required for the uptake of each form of tau.

## Uptake of monomeric and fibrillar tau uses different receptors but similar endocytic and intracellular trafficking pathways

The functional genomics screen analyses suggest that, although the early steps of tau endocytosis differ for monomeric and fibrillar tau, including surface receptors, the later stages show considerable convergence. Confirming this, inhibition of dynamin-dependent endocytosis greatly reduced uptake of both monomeric and fibrillar forms of tau (Figs. 4A,B and EV4). Vacuolar-type ATPases have a number of roles in the endolysosomal system, including modulating clathrin-mediated endocytosis (Kozik et al, 2013) and regulating acidification of late endosomes and lysosomes (Collins and Forgac, 2020). Therefore, tau uptake by neurons was assessed following treatment with bafilomycin A, the V-type ATPase inhibitor (Bowman et al, 1988). Human neurons treated with bafilomycin A before the addition of extracellular tau-pHrodo were markedly reduced in detectable intracellular pHrodo-tau. Similarly, loss of three of the subunits of complex II of class III PI3K all lead to reduced tau uptake, suggesting that class III PI3K activity is required for uptake of all forms of tau, which was confirmed by inhibiting PI3K activity with PIK93 or wortmannin (Cuenda and Alessi, 2000) administration before addition of extracellular monomeric or fibrillar tau (Fig. 4A,B).

To test whether disruption of trafficking from the Golgi impacts neuronal ability to endocytose extracellular tau, human iPSC-derived excitatory neurons were treated with Brefeldin A. Live imaging of pHrodo-tau uptake demonstrated reduced uptake of extracellular monomeric and fibrillar tau (Figs. 4C,D and EV4), suggesting that uptake of both monomeric and fibrillar tau is

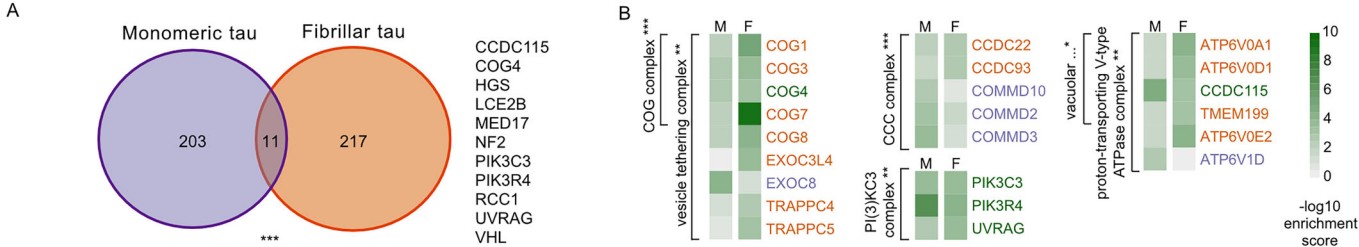

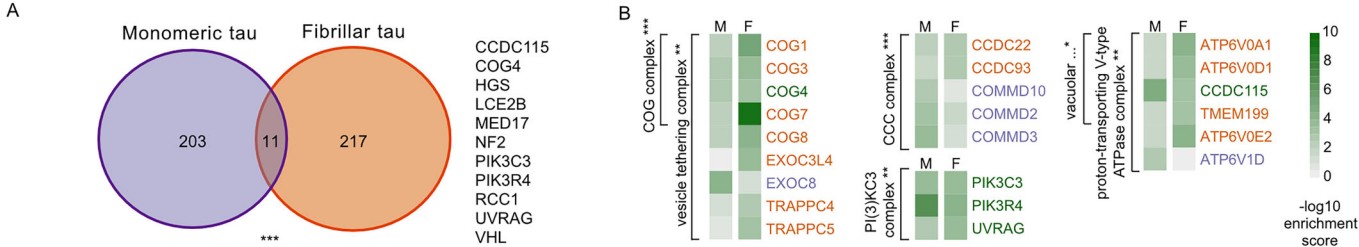

**Figure 3. Uptake mechanisms of monomeric and fibrillar tau differ at the neuronal surface but share intracellular trafficking pathways.**

(A) Monomeric and fibrillar tau share common intracellular pathways for neuronal entry. Genes identified as required for monomeric or fibrillar tau uptake are significantly overlapping (hypergeometric test, *p* < 0.001). The 11 genes identified in both screens are shown on the right side. (B) Subunits of protein complexes that are significantly enriched in the combined set of genes required for uptake of either form of tau. The colour indicates the positive enrichment score in the monomeric (M) and fibrillar (F) screens, respectively. Only subunits identified as required in either screen are shown (orange if identified in fibrillar screens, purple if identified in monomeric screens, bold green if identified in both). FDR <0.05 (*), <0.01 (**), <0.001 (***), with significance determined by a hypergeometric test implemented in gProfiler2 (see methods for details). C Positions of genes identified as significantly enriched in our CRISPR screens in the cellular compartments associated with the gene-encoded proteins, using the COMPARTMENTS dataset (FDR <0.05). Genes influencing uptake of monomeric (purple), fibrillar (orange) or both (green) forms of tau are ordered using the positive enrichment ranking for each cellular compartment, where appropriate genes appear in multiple compartments. See also Fig. EV3; Dataset EV3.

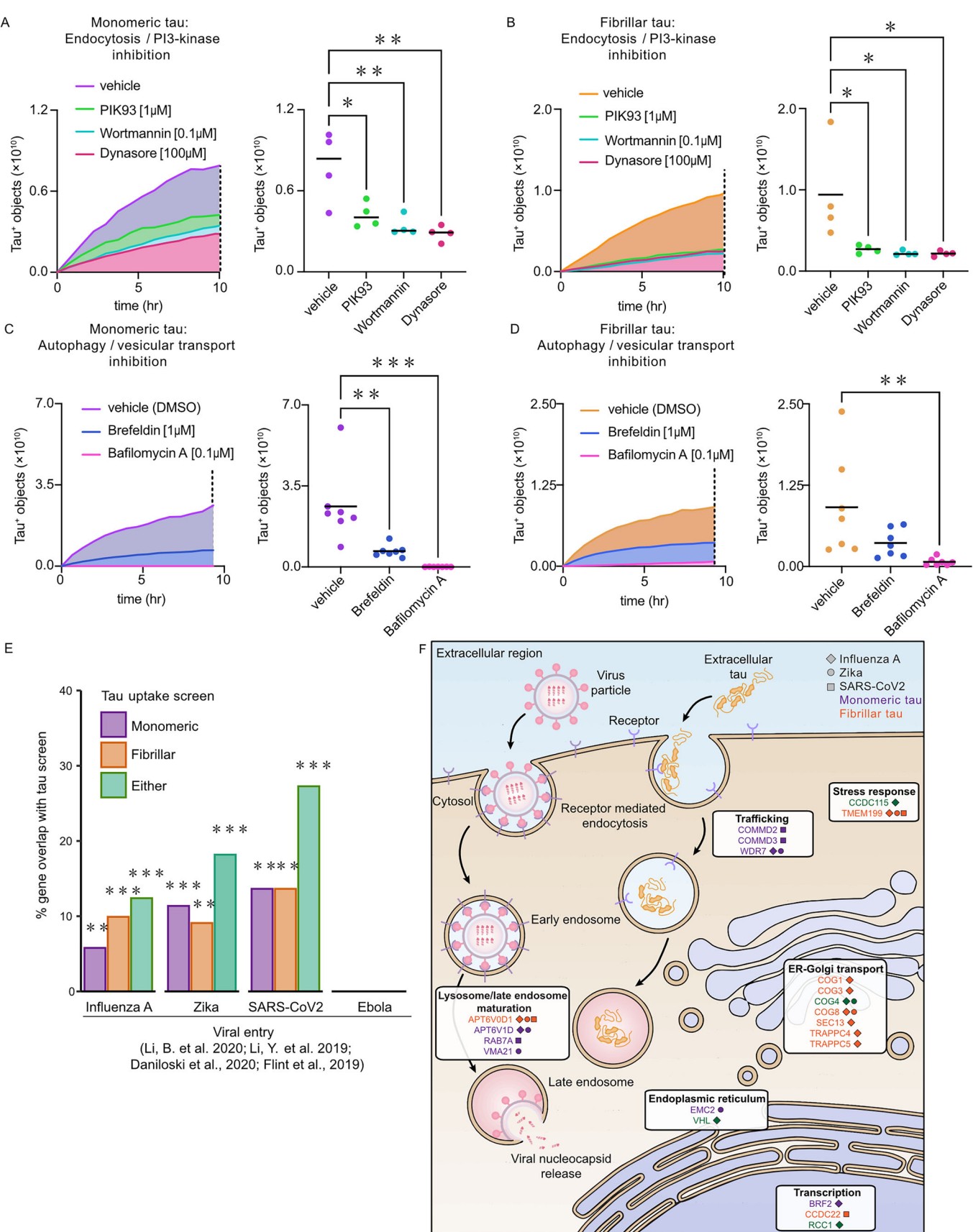

◀ **Figure 4. Neuronal uptake of extracellular tau shares functional similarities with endocytic pathways involved in viral entry.**

(A, B) Effect of endocytosis and PI3-kinase activity inhibition on neuronal uptake of extracellular monomeric (A) and fibrillar tau-pHrodo (B). Neurons were pre-incubated with either 100 μM Dynasore, 1 μM PIK93, 0.1 μM wortmannin or vehicle control (0.1% [v/v] DMSO) for 1 week, before addition of tau-pHrodo. pHrodo-positive objects were counted over 10 h at 45-min intervals. Object measurements are displayed over time and scatter plots of the final time point (dashed line) are shown; bars indicate mean. Four wells per treatment, and at least 20 fields of view per well. (C, D) Effect of autophagy and vesicular transport inhibition on neuronal uptake of extracellular monomeric (C) and fibrillar tau-pHrodo (C) into low-pH compartments. Neurons were pre-incubated for 24 h with either 1 μM Brefeldin, 0.1 μM Bafilomycin A or vehicle control (0.1% [v/v] DMSO) using the same experimental conditions and parameters as in (A). The number of pHrodo-positive objects was measured as in (E-F; seven wells per treatment). Statistical significance was determined using one-way ANOVA, Dunnett's test for multiple comparisons (monomeric/fibrillar tau uptake with 100 μM Dynasore **$p = 0.0013$/ *$p = 0.00153$; 1 μM PIK93 *$p = 0.0128$/*$p = 0.0239$; 0.1 μM wortmannin **$p = 0.0031$/ *$p = 0.016$; 1 μM Brefeldin **$p = 0.0023$/ns; 0.1 μM Bafilomycin A ***$p = 0.0001$/**$p = 0.0067$). (E) The sets of genes required for human neuronal uptake of monomeric, fibrillar or either form of tau significantly overlap with the sets of genes required for receptor-mediated endocytosis of Influenza A, Zika and SARS-CoV-2 viruses, but not macropinocytosis of Ebola virus. Gene sets required for entry were defined from their respective screens using a threshold of FDR <0.05, and hypergeometric tests were used to evaluate the significance of overlap between gene sets (FDR-corrected *$p$* < 0.05, *$p$** < 0.01, *$p$*** < 0.001). (F) Diagram of cellular compartments showing genes shared between tau uptake (FDR <0.05) and viral entry screens and their predicted roles in tau uptake and virus life cycle. Genes are marked by a shape corresponding to Influenza A virus (diamond), Zika (circle) or SARS-CoV-2 virus entry (square) and coloured according to monomeric (purple), fibrillar (orange) or either form of tau (green). See also Fig. EV4; Dataset EV4.

dependent on the integrity of ER-Golgi transport. However, given the widespread disruption of intracellular vesicular trafficking that is induced by Brefeldin A treatment, it may be that reduced tau uptake in this case is due to secondary effects on endolysosome function.

Current models for tau-mediated propagation of pathology between neurons propose pathogenic forms of tau seed aggregation in receiving cells in the cytoplasm. Thus, following endocytosis, extracellular tau is likely to exit the endolysosome as part of its pathogenic mechanism. This echoes the life cycle of many viruses, which use a variety of mechanisms to enter human cells, including receptor-mediated endocytosis and micropinocytosis (Thorley et al, 2010). After escaping membrane-bound compartments, viral genetic material can be delivered to the cytoplasm or nucleus for replication. The host factors or genes that regulate the entry and life cycle of several viruses have been comprehensively studied using RNAi and CRISPR screens. These include the neurotropic Zika (Li et al, 2019), Influenza A (Li et al, 2020) and SARS-CoV-2 (Daniloski et al, 2021) viruses, which enter cells via receptor-mediated endocytosis, and Ebola (Flint et al, 2019), which uses micropinocytosis.

To compare the biology of tau entry and processing with that of genes required for different viruses entry, the sets of genes identified in the screens here were compared with those reported for Influenza A, Zika, SARS-CoV2 and Ebola infection (Fig. 4E). Genes required for the uptake of tau were also compared to genes required for transferrin or epidermal growth factor (EGF) that enter via endocytosis (Collinet et al, 2010) or phagocytosis of beads of different size and charge (Haney et al, 2018) (Fig. EV4). Significant overlaps in the number of genes identified in either monomeric and fibrillar tau screens were found with those reported for Influenza A, Zika and SARS-CoV-2 infection, but not with Ebola infection (Fig. 4E) or those required for phagocytosis of beads (Fig. EV4). Monomeric tau uptake shows a small but significant overlap with the endocytosis of transferrin, but there were no other significant overlaps between fibrillar tau uptake, transferrin or EGF endocytosis (Fig. EV4).

Genes required for both tau entry and either Influenza A, Zika or SARS-CoV2 infection and replication are significantly enriched in genes encoding the vATPase or its assembly, the COG complex and intracellular vesicular trafficking, as reflected in the gene

ontology enrichments (Fig. EV4), with their encoded proteins enriched in proteins localised to the endosome, lysosome and Golgi network (Figs. 4F and EV4). The overlap in routes of cell entry by viruses and tau protein, at the levels of both individual genes and cellular pathways, suggests that the cellular factors that regulate tau entry to neurons are functionally similar to the host factors regulating the intracellular viral life cycle, including cellular entry and access to the cytoplasm.

## LRRK2 gain and loss of function mutations disrupt the neuronal endolysosomal system

The large, multidomain protein encoded by *LRRK2* has multiple roles in endocytosis and vesicle trafficking (Taylor and Alessi, 2020) and mutations in *LRRK2* are causal for autosomal dominant PD with neuronal tau aggregation (Rajput et al, 2006; Pouloupoulos et al, 2012). LRRK2 has been found to regulate endolysosomal protein trafficking through its interaction with AP2 (Roosen and Cookson, 2016; Heaton et al, 2020; Liu et al, 2021). Mutations in the kinase domain of *LRRK2* or in domains that regulate kinase activity are causal for PD (Henderson et al, 2019). The most prevalent LRRK2 PD mutation, LRRK2 G2019S (Gilks et al, 2005), stimulates kinase activity (West et al, 2005) and disrupts the endolysosomal system in a variety of transgenic and cell models (Taylor and Alessi, 2020).

We generated cortical neurons from isogenic *LRRK2* wild type, *LRRK2* heterozygous and homozygous null and LRRK2 G2019S iPSCs (Figs. 5A,B and EV5). Gene expression analysis confirmed that cultures of each genotype contained excitatory and inhibitory interneurons, with one LRRK2 null and the LRRK2 G2019S mutation iPSCs generating proportionally more interneurons, based on expression of genes indicative of each cell type (Fig. EV5). Assessment of the endolysosomal system in *LRRK2* mutant neurons found that *LRRK2* homozygous null neurons have accumulation of LAMP1-positive late endosomes and lysosomes, as previously described in other models (Henry et al, 2015; Pellegrini et al, 2018), compared with isogenic controls and LRRK2 heterozygotes, whereas *LRRK2* heterozygous nulls do not have measurable defects (Fig. 5C,D). The endosomal compartments of the LRRK2 null and isogenic control neurons did not show detectable changes (Fig. 5). LRRK2 G2019S mutant neurons also

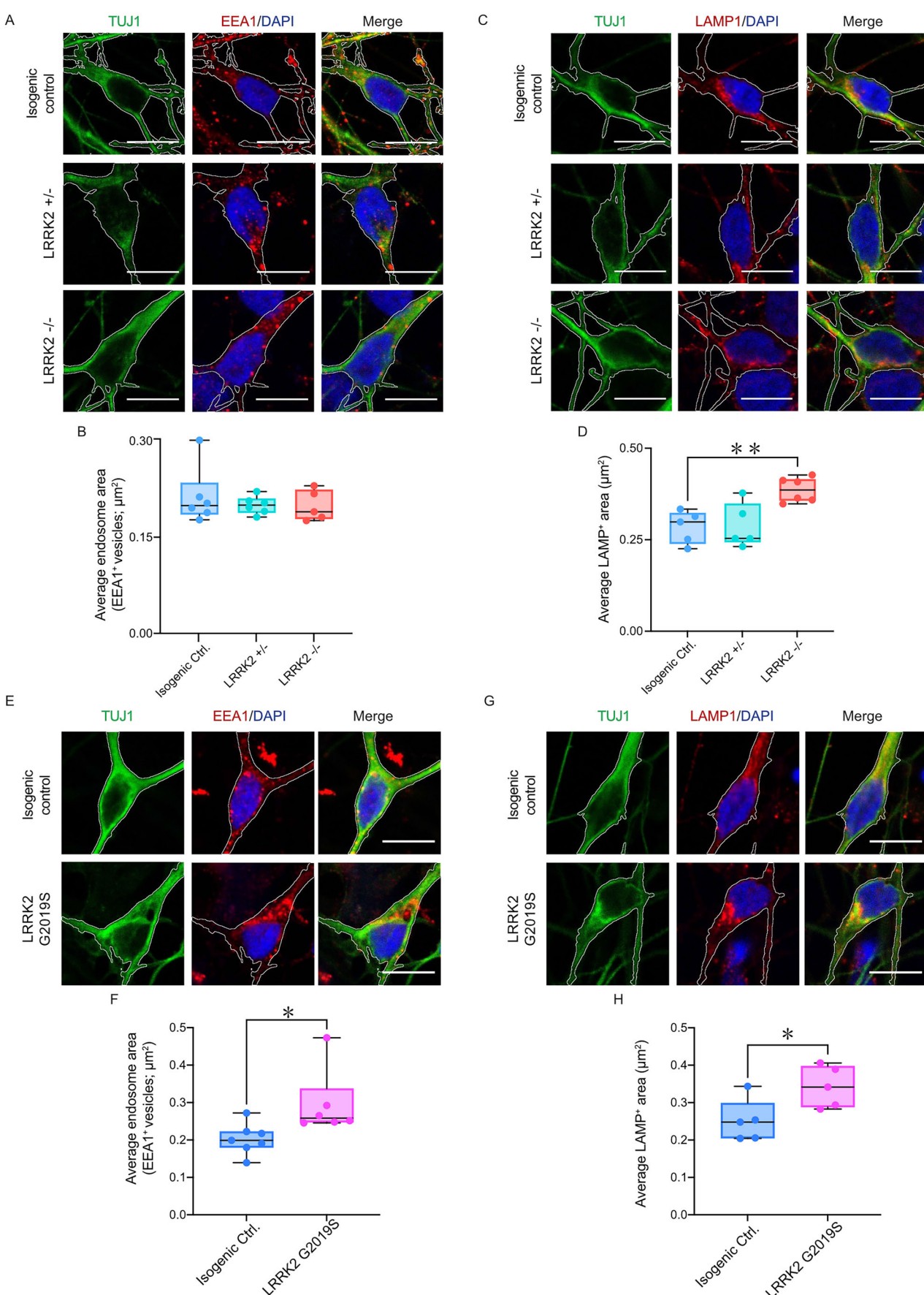

**Figure 5.   LRRK2 gain and loss of function mutations affect the endolysosomal system in excitatory neurons.**

(A) LRRK2 heterozygous and homozygous knockout neurons do not exhibit an endosomal phenotype. Representative immunohistochemistry of neurons from isogenic control, heterozygous (+/−) and homozygous (−/−) LRRK2 knockout human iPSCs (60 days after induction), immunostained for neuronal β3-tubulin (TUJ1, green), early endosomes (EEA1, red) and nuclei were counterstained (DAPI, blue). Scale bar, 10 μm. (B) No significant changes in the average size of early endosomes (EEA positive vesicles; μm²) in heterozygous and homozygous (null) LRRK2 knockout neurons compared with isogenic control (n = >5 images, one-way ANOVA, Dunnett's test for multiple comparisons). Box plot line represents median, box covers the interquartile interval and whiskers represent min to max. (C) LRRK2 homozygous knockout neurons display accumulation of LAMP1+ vesicles, including late endosomes, lysosomes and autolysosomes. Representative immunohistochemistry of neurons from isogenic control, heterozygous (+/−) and homozygous (−/−) LRRK2 knockout human iPSCs (60 days after induction), immunostained for neuronal β3-tubulin (TUJ1, green), LAMP1 (red) and nuclei were counterstained (DAPI, blue). Scale bar, 10 μm. (D) Significant increase in average late endosome/lysosome/autolysosome area (LAMP1 positive vesicles; μm²) in homozygous LRRK2 knockout neurons compared with isogenic controls (n = >5 images; one-way ANOVA, Dunnett's test for multiple comparisons; **p = 0.006). Box plot line represents median, box covers the interquartile interval and whiskers min to max. (E–H) Neurons heterozygous for the LRRK2 G2019S mutation display accumulation of LAMP1+ vesicles, including late endosomes, lysosomes and autolysosomes. Representative confocal images of human iPSC neurons from isogenic control and with LRRK2 G2019S mutation (heterozygous; 60 days after induction), immunostained for neuronal β3-tubulin (TUJ1, green), (E) endosomes (EEA1, red) or (G) LAMP1 (red) and nuclei were counterstained (DAPI, blue). Scale bar, 10 μm. Significant increase in average area of early (F; Student's t-test *p = 0.0298) and late endosomes/lysosomes (H) (EEA1/LAMP1 positive vesicles; μm²) in LRRK2 G2019S neurons compared with isogenic controls (n = >5 images). Box plot line represents median, box covers the interquartile interval and whiskers represent min to max.

showed clear accumulation of early endosome and late endosome/lysosomes, compared with isogenic controls (Fig. 5), as has previously been reported in transgenic LRRK2 G2019S mouse astrocytes (Henry et al, 2015). Thus, complete loss of LRRK2 function and kinase-activating mutations in LRRK2 both result in abnormalities of the neuronal endolysosomal system, which are more severe in LRRK2 G2019S neurons than in LRRK2-null neurons, extending to early endosomes (Fig. 5E–H).

## LRRK2 regulates neuronal endocytosis of monomeric tau, alpha-synuclein and Abeta

To investigate the role of *LRRK2* in neuronal endocytosis of extracellular tau, uptake of extracellular monomeric and fibrillar tau by *LRRK2* heterozygous and homozygous null and *LRRK2* G2019S neurons was analysed by flow cytometry (Figs. 6A,B and EV6). *LRRK2* null neurons were significantly reduced in their ability to take up monomeric tau (both wild-type and tau P301S forms), while remaining competent for transferrin uptake (Fig. EV6). Fibrillar tau uptake was not affected by loss of the *LRRK2* gene, confirming the findings of the primary CRISPR screens (Fig. 6C). In contrast, *LRRK2* G2019S neurons demonstrated increased uptake of both monomeric and fibrillar forms of tau (Figs. 6 and EV6).

Given that both loss of LRRK2 function and the kinase-activating LRRK2 G2019S mutation disrupt the endolysosomal system, uptake of additional cargoes relevant to neurodegenerative disease were analysed in the different LRRK2 genotypes (Fig. 6D–F). Uptake of monomeric alpha-synuclein, which is also mediated by LRP1 (Chen et al, 2022), was altered in LRRK2 null and G2019S neurons in the same manner as tau, with loss of function reducing uptake, and the G2019S mutation increasing uptake (Fig. 6D). Loss of LRRK2 function had no impact on Abeta peptide uptake, whereas the G2019S mutation reduced Abeta uptake (Fig. 6E). Finally, receptor-mediated endocytosis of EGF was also reduced by the LRRK2 G2019S mutation, and increased by loss of LRRK2 function (Figs. 6F and EV6).

Therefore, we conclude that LRRK2 affects the uptake of different cargoes that enter neurons via distinct routes, by regulating multiple endocytic pathways. In particular, tau receptor LRP1-mediated endocytosis pathways are seemingly attenuated by loss of LRRK2 function, as indicated by an accumulation of LRP1 on the cell surface of LRRK2 null neurons (Fig. EV6).

## Inhibition of the LRRK2 kinase broadly perturbs receptor-mediated endocytosis

Enhanced LRRK2 kinase activity due to the *LRRK2 G2019S* mutation and reduction in LRRK2 protein levels both dysregulate the neuronal endolysosomal system, but alter tau uptake in different ways. The *LRRK2 G2019S* mutation results in increased uptake of monomeric and fibrillar tau, whereas complete loss of LRRK2 selectively reduces uptake of monomeric, and not fibrillar tau. To further test the dependency of tau uptake on LRRK2 kinase activity, wild-type iPSC-derived neurons were treated with small molecule LRRK2 kinase inhibitors for one week before testing uptake of extracellular monomeric or fibrillar tau (Fig. 7). Following treatment of neurons with LRRK2 selective kinase inhibitors MLi-2 (Fell et al, 2015) and GSK3357679A (Singh et al, 2021; Tasegian et al, 2021), no significant differences in the size of endosomes, or late endosomes/lysosomes were detected (Figs. 7A,B and EV7). Neuronal uptake of pHrodo-labelled monomeric and fibrillar tau, measured using live-imaging, was reduced after treatment with LRRK2 inhibitors (Fig. 7C,D). Neuronal uptake of transferrin (Alexa 633 conjugate) was not significantly affected by LRRK2 inhibitors (Fig. 7E).

To examine if the apparent LRRK2 kinase inhibition affected neuronal surface levels of LRP1, neurons were treated with MLi-2 or GSK3357679A for 1 week, and surface proteins were labelled with biotin, following the same approach as with the LRRK2 genotypes (Fig. EV6). Neuronal LRP1 surface levels were increased after treatment with GSK3357679A, compared with vehicle-treated controls. However, this effect was not observed in neurons treated with MLi-2, a difference that may reflect differences in potency between the two inhibitors (Fig. EV7) (Tasegian et al, 2021). These results highlight the requirement for LRRK2 kinase activity in both monomeric and fibrillar tau uptake, and underscore the differing impacts on tau endocytosis of modulation of LRRK2 kinase activity and reduction or loss of LRRK2 protein.

## Discussion

We report here the delineation by whole-genome CRISPR knock-out screens of the genes required for extracellular tau uptake by human cortical excitatory neurons, the primary cell type affected in tauopathies. Independent screens for full-length monomeric and synthetic fibrillar P301S tau avoided making assumptions about the

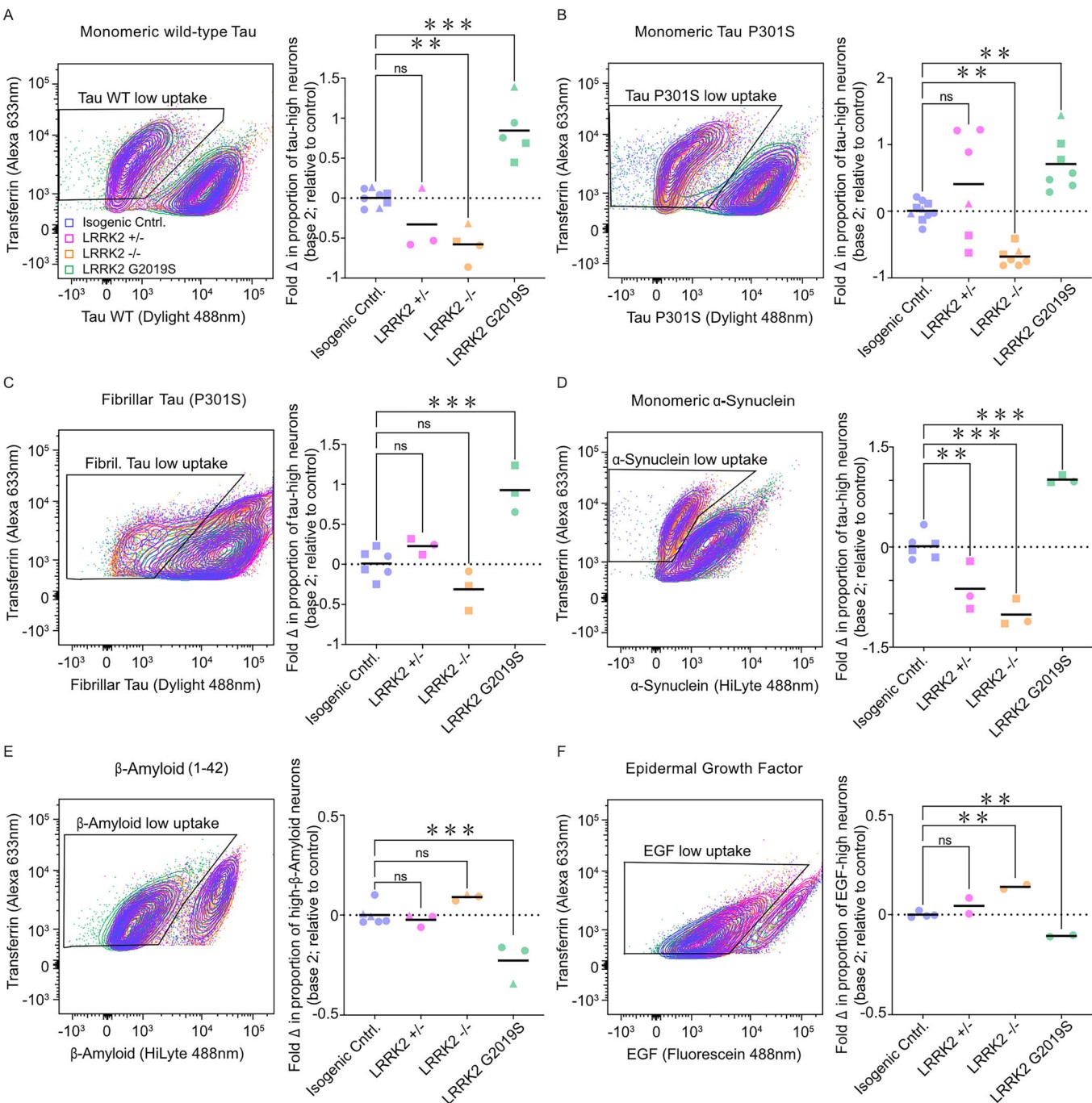

**Figure 6. LRRK2 gain and loss of function mutations affect uptake of extracellular tau, a-synuclein and a-beta.**

Neurons with LRRK2 gain and loss of function mutations differ in their ability to take up extracellular proteins involved in neurodegeneration. Isogenic control, LRRK2 heterozygous (−/+), homozygous (−/−) null or LRRK2 G2019S mutant neurons (65 days after induction) were incubated with 90 nM transferrin (Tfn) Alexa 633 nm and either monomeric wild-type (**A**), monomeric P301S (**B**) or fibrillar P301S (**C**) tau Dylight 488 nm for 4 h before dissociation into single cells and analysis by flow cytometry. Neurons from each of the genotypes were also incubated with α-synuclein (**D**), β-amyloid (1–42) (**E**) HiLyte 488 nm or epidermal growth factor (EGF) Fluorescein 488 nm (**F**) and analysed in the same way as the tau protein derivatives. For each genotype and cell line indicated, the percentage of low protein uptake in transferrin-positive neurons (indicated by the black outlined boxes) was compared to the mean (dashed line) of protein uptake of control neurons and displayed as fold change (log2). Significance was determined using one-way ANOVA with Dunnett's test for multiple comparisons (monomeric tau WT: LRRK2 [−/− **$p$ = 0.0053; G2019S ***$p$ < 0.0001; monomeric tau P301S: LRRK2 [−/− **$p$ = 0.0076; G2019S **$p$ = 0.0064]; fibrillar tau P301S: LRRK2 G2019S ***$p$ = 0.0002; α-synuclein: LRRK2 [+/− **$p$ = 0.0056; −/− ***$p$ = 0.0001; G2019S ***$p$ = 0.0002]; β-amyloid: LRRK2 G2019S ***$p$ = 0.0004]; EGF: LRRK2 [−/− **$p$ = 0.0022; G2019S **$p$ = 0.0086]). Circles, triangles and squares represent independent experiments; $n$ = >3). See also Fig. EV6.

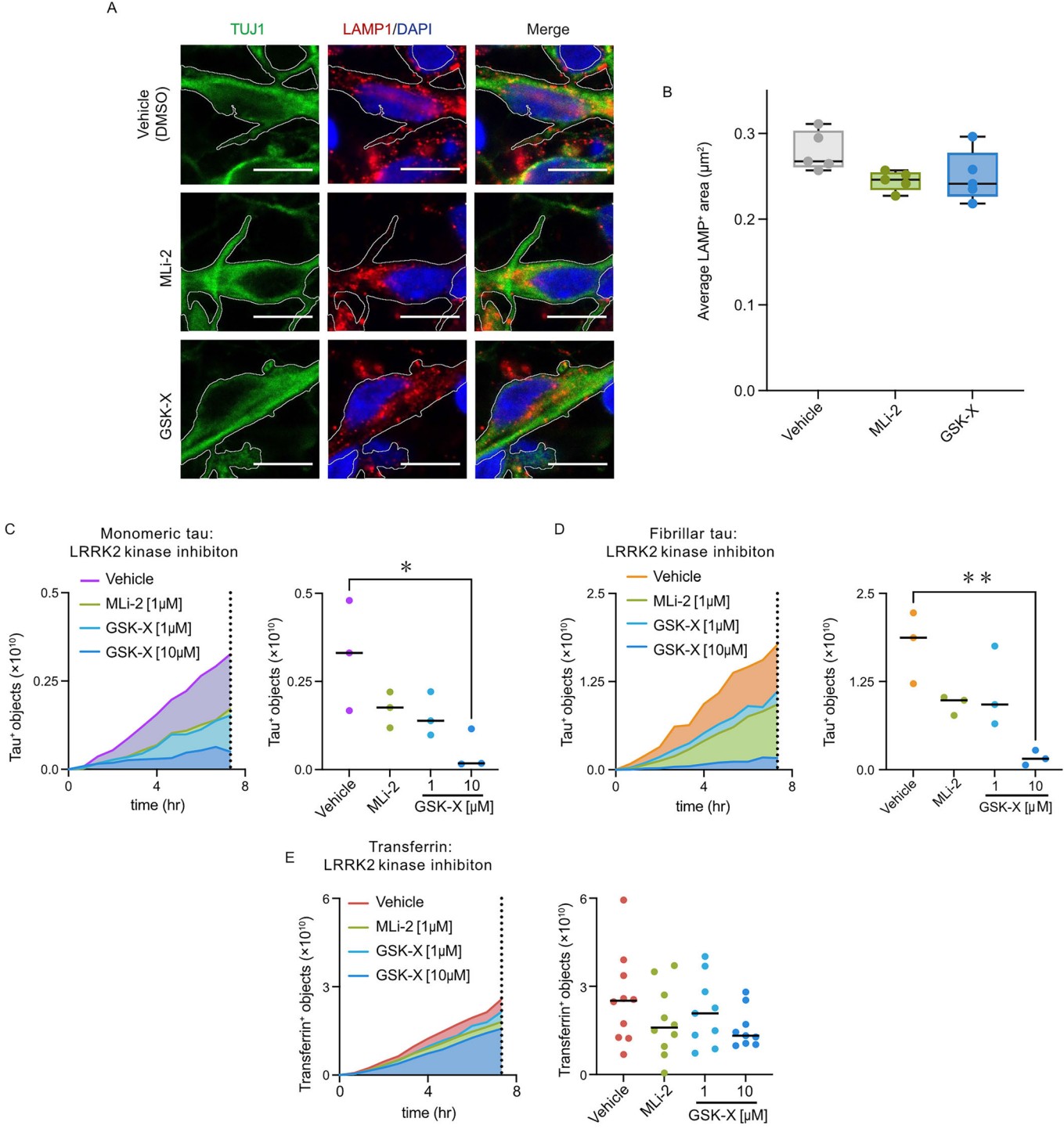

forms and fragments of tau that are present in interstitial fluid in the CNS and of the forms that are competent to enter neurons, providing a broad view of the mechanisms for tau entry to human neurons. Measurement of tau uptake was restricted to analysing the amount of labelled, extracellular tau that was present in neurons hours after uptake, limiting the effect of intracellular degradation of tau. Consistent with our previous findings (Evans et al, 2018), whole genome screens found that monomeric and fibrillar tau are taken up by neurons by overlapping pathways, with significant differences in the initial steps of endocytosis, most notably the receptors used, the dependency of fibrillar tau uptake on protein glycosylation and the requirement for LRRK2 for monomeric tau endocytosis. The major mode of entry identified in this study for both monomeric and fibrillar tau is endocytosis mediated by different receptors, which is perturbed by a variety of disruptions of the integrity of the endolysosome-autophagy system and

**Figure 7.  Inhibition of the LRRK2 kinase reduces neuronal uptake of monomeric and fibrillar tau.**

(A, B) Neurons treated with LRRK2 kinase activity inhibitors do not exhibit accumulation of LAMP1+ vesicles (late endosomes, lysosomes and autolysosomes). Representative confocal images of neurons (60 days after induction) treated for 1 week with vehicle control (0.1% [v/v] DMSO), 1 μM MLi-2 or 10 μM GSK3357679A (GSK-X), immunostained for neuronal β3-tubulin (TUJ1, green), LAMP1 (red) and nuclei were counterstained (DAPI, blue). Scale bar, 10 μm. No significant changes were detected in the average area of LAMP1+ structures (μm$^2$) in neurons treated with LRRK2 kinase activity inhibitors compared with vehicle control (n = >5 images). Box plot line represents median, box covers the interquartile interval and whiskers represent min to max. (C–E) Entry of monomeric and fibrillar tau into neurons is inhibited by LRRK2 kinase inhibitors (MLi-2 and GSK-X), in a concentration-dependent manner. Human iPSC-derived neurons were pre-incubated with 1 μM MLi-2, 1 μM or 10 μM GSK-X or vehicle control (0.1% [v/v] DMSO) for one week, before acute addition of pHrodo-labelled monomeric tau (C), fibrillar tau (D) or transferrin (E). Live-imaging captured pHrodo-labelled protein uptake into low-pH compartments in neurons (65 days after induction), and the number of pHrodo-positive objects was measured over 7.3 h at 45-min intervals. Object measurements are displayed over time, and scatter plots of the final time point (dashed line) are shown; bars indicate the mean. At least three wells per treatment, and >20 fields of view per well. Significance was determined using one-way ANOVA with Dunnett's test for multiple comparisons (monomeric/ fibrillar tau with 10 μM GSK-X *p = 0.0161/**p = 0.0082). See also Fig. EV7.

intracellular vesicular transport between the Golgi and other cellular compartments.

This study confirmed that the low-density lipoprotein receptor LRP1 is a primary receptor for monomeric tau in human neurons, as recently described (Rauch et al, 2020). The identification of *LRP1* as one of the top two-ranked genes required for monomeric tau uptake is an important validation of the functional genomics approach taken here to understand the cell biology of neuronal tau uptake. It is noteworthy that two different strategies for acutely inhibiting tau-LRP1 interactions, using the RAP chaperone and domain IV of LRP1, confirmed that LRP1 mediates the majority of monomeric tau uptake in neurons, but is not a major receptor for fibrillar tau. This extends to human AD brain fibrillar tau, which we found does efficiently enter human neurons, but does not require interacting with LRP1 to do so.

Although these screens did not identify a single major receptor for fibrillar tau, analogous to LRP1 for monomeric and oligomeric tau, the screens confirmed the established link between heparan sulfate proteoglycans and fibrillar tau uptake (Holmes et al, 2014). Tau uptake has previously been shown to be mediated by plasma membrane heparan sulfate proteoglycans (Holmes et al, 2014), and mutations in COG complex components result in congenital disorders of glycosylation in humans that have a wide spectrum of neurodevelopmental pathologies (D'Souza et al, 2020). Two genes required for fibrillar tau uptake, *B3GALNT1* and *B3GNTL1*, encode glycosyltransferases (Ricci Hagman et al, 2018), and the cell surface proteoglycan, *GPC5*, identified in Alzheimer's disease GWAS (Wang et al, 2020), is also required for fibrillar tau entry. Thus, it is likely that the loss of COGs and the glycosyltransferases both result in disruption of cell surface heparan sulfate proteoglycan assembly and thus reduced fibrillar tau uptake.

Beyond the plasma membrane and the initial steps of endocytosis, CRISPR screens for uptake of monomeric and fibrillar tau identified genes and pathways required for uptake of both forms of the protein, but with some differences. Although the specific genes identified in each screen were not identical, there was a high degree of convergence of hits in protein complexes, organelles and cellular pathways. For each form of tau, key regulators of receptor-mediated endocytosis were identified at the level of endocytosis for monomeric tau (clathrin heavy chain and dynamin-2), and within early endosomes for fibrillar tau (EEA1). Beyond the early endosome, tau uptake by human neurons is highly dependent on class III PI3-kinase, which was confirmed by acute small-molecule inhibition. Class III PI3K is a major regulator of both endosome recycling to the plasma membrane, as well as

autophagy and phagocytosis (Bilanges et al, 2019), and its involvement highlights that perturbations of receptor recycling and of wider vesicular trafficking, including autophagy, result in a loss of neuronal capacity for tau uptake. Therefore, we conclude that monomeric/oligomeric and fibrillar tau use different receptors and early endocytic pathways to enter neurons, which then converge in the late endosome/lysosome.

Alterations in receptor recycling are a well-characterised consequence of the loss of function of proteins involved in endosome sorting and recycling (Cullen and Steinberg, 2018). We found that loss of components of the CCC complex reduces neuronal uptake of fibrillar and monomeric tau. Sorting nexin-17 (SNX17) is required for monomeric tau uptake, and given that a key cargo for SNX17-mediated recycling is LRP1 and SNX17 interacts with the WASH/CCC complex (McNally et al, 2017), it is likely that loss of each of these proteins affects plasma membrane levels of LRP1. Similarly, late endosome and lysosome acidification is essential for receptor recycling during the sorting process (McNally and Cullen, 2018), and we find that loss of multiple vATPase subunits, as well as accessory subunits and assembly proteins, all reduce tau uptake.

The dependency of tau uptake on surface receptor expression, receptor-mediated endocytosis and wider homoeostasis of intracellular vesicular trafficking has similarities with the host factors regulating viral entry and replication, particularly for viruses that enter human cells by receptor-mediated endocytosis. Comparing the results of this study with loss-of-function screens for viral entry, we find significant overlap in genes and protein complexes required in each case. This ranges from endocytosis, through vATPase activity, to the COG and TRAPPC complexes in the Golgi. The similarities in the cell biology of the two processes indicate that tau entry to neurons is a quasi-infectious process, which adapts normal cellular processes to both gain entry to the endolysosome and to propagate within neurons.

We also report here that LRRK2 is required for the uptake of monomeric, but not fibrillar tau, by human neurons. In contrast, we found that neurons expressing the LRRK2 G2019S, kinase-activating mutation have increased uptake of both extracellular monomeric and fibrillar tau, together with disruption of the endolysosomal system. Consistent with the LRRK2 G2019S gain-of-function data, acute inhibition of the kinase activity of LRRK2 inhibits neuronal uptake of all forms of tau without detectable significant morphological disruption of the endolysosomal system, within the detection limits of the number of replicate experiments and imaging systems used here. The role of LRRK2 in endocytosis

is not limited to tau, as it also regulates uptake of alpha-synuclein, another LRP1 ligand (Chen et al, 2022). The mechanisms by which LRRK2 regulates tau and synuclein endocytosis remain to be elucidated, and may be a direct effect on receptor-mediated endocytosis through LRRK2 interaction with the AP2 clathrin adaptor complex (Roosen and Cookson, 2016; Heaton et al, 2020; Liu et al, 2021). Alternatively, given the widespread changes in the endolysosomal-autophagy system downstream of LRRK2 gain- and loss-of-function (Taylor and Alessi, 2020), LRRK2 may indirectly impact endocytosis of certain cargoes, secondary to perturbation of the endolysosome.

The involvement of LRRK2 and LRP1 in both tau and synuclein uptake by human neurons addresses questions about the role of LRRK2 in tauopathies and the mechanistic links between the pathogenesis of PD and AD and other tauopathies, while opening some new ones. Missense, dominant mutations in *LRRK2* are a cause of familial PD and result in kinase activation, and LRRK2 variants are risk factors for sporadic PD (Hardy, 2010). *LRRK2* mutations causative for genetic PD are commonly accompanied by intraneuronal tau aggregation in the cerebral cortex (Henderson et al, 2019). Furthermore, missense *MAPT* mutations cause frontotemporal dementia with parkinsonism, and GWAS have identified *MAPT*

variants as risk factors for PD (Hardy, 2010). Thus, both *LRRK2* and *MAPT* are associated with the pathogenesis of PD, with *LRRK2* mutations facilitating the development of tau pathology in PD and PD dementia. LRP1 is a receptor for neuronal endocytosis of both tau and alpha-synuclein (Rauch et al, 2020; Chen et al, 2022), and we report here that LRRK2 regulates endocytosis of both tau and alpha-synuclein by human neurons. Recent GWAS investigating the rate of progression of a tauopathy in progressive supranuclear palsy identified variants in a putative enhancer for *LRRK2* that increase the rate of progression, implicating LRRK2 in mediating tauopathy pathogenesis (Jabbari et al, 2021). Together, these findings place LRRK2 together with LRP1 as contributing to tauopathy pathogenesis, and suggest that small molecule inhibition of LRRK2, or reduction in LRRK2 protein levels, may have roles in slowing disease progression in neurodegenerative diseases involving spread of tau aggregation, including Alzheimer's disease, as well as PD progression mediated by alpha-synuclein.

# Methods

## Reagents and tools table

| Reagent/resource | Reference or source | Identifier or catalogue number |
| --- | --- | --- |
| **Experimental models** | | |
| KOLF2-C1 | Welcome Sanger Institute | WTSIi018-B-1 |
| KOLF2-C1 Cas9 | Andrew Basset lab; Welcome Sanger Institute | |
| KOLF2-C1 LRRK2 −/+ | Andrew Basset lab; Welcome Sanger Institute | |
| KOLF2-C1 LRRK2 −/− | Andrew Basset lab; Welcome Sanger Institute | |
| KOLF2-C1 LRRK2 G2019S −/+ | Andrew Basset lab; Welcome Sanger Institute | |
| HEK 293 cells | | |
| **Recombinant DNA** | | |
| pAAV-Neo_CAG-Cas9 | Addgene | #86698 |
| pKLV2-U6gRNA(gGFP)-PGKmCherry2AGFP-W | Addgene | #67982 |
| Human CRISPR Library v.1.1 | Addgene | #67989 |
| pET24d_P301S_FLAG×3_10×his-tag | Evans et al, 2018 | |
| pET24d_FLAG×3_10×his-tag | Evans et al, 2018 | |
| pKLV2-U6gRNA5(BbsI)-ccdb-PGKpuro2ABFP-W | Addgene | # 67974 |
| **Antibodies** | | |
| tau | Dako | # A0024 |
| MAP2 | abcam | #ab5392 |
| Tuj1 | Cambridge Bioscience; | #ab14545 |
| TBR1 (Abcam) | abcam | #ab31940 |
| CTIP2 (Abcam) | abcam | #ab18465 |
| EEA1 (Abcam) | abcam | #ab109110 |
| LAMP1 (Abcam) | abcam | #ab62562 |
| Secondary antibody anti-mouse Alexa 647 | Thermo Fisher Scientific | #A32787 |
| Secondary antibody anti-rat Alexa 647 | Thermo Fisher Scientific | #A48265 |
| Secondary antibody goat anti-chicken Alexa 488 | Thermo Fisher Scientific | #A11039 |
| Secondary antibody goat anti-rabbit Alexa 546 | Thermo Fisher Scientific | #A11010 |

| Reagent/resource | Reference or source | Identifier or catalogue number |
|---|---|---|
| Milli-Mark Anti-MAP2-PE antibody | Sigma, Merck | # FCMAB318PE |
| LRP1 | Abcam | # ab92544 |
| N-Cadherin | Abcam | # ab18203 |
| β-Actin | Sigma, Merck | # A2228 |
| Secondary IgG antibody Alexa Fluor 488 | Thermo Fisher Scientific | # A-2120 |
| Secondary IgG antibody Alexa Fluor 546 | Thermo Fisher Scientific | # A10040 |
| **Oligonucleotides and other sequence-based reagents** | | |
| AAVS1 locus target: ggggccactagggacaggattgg | Andrew Basset lab; Welcome Sanger Institute | |
| LRRK2-targeting gRNA sequence: ATTGCAAAGATTGCTGACTA | Synthego | |
| LRRK2 G2019S homology-directed repair template: TTTCACACTGTATCCCAATGCTGCCATCATTGCAAA GATTGCTGACTACAGCATTGCTCAGTACTGCTGTA GAATGGGGATAAAAACATCAGAGGGCACAC | Andrew Basset lab; Welcome Sanger Institute | |
| high-throughput sequencing of LRRK2 KO F - ACACTCTTTCCCTACACGACGCTCTTCCGATCT cagatacctccactcagcca | Andrew Basset lab; Welcome Sanger Institute | |
| high-throughput sequencing of LRRK2 KO R - TCGGCATTCCTGCTGAACCGCTCTTCCGAT CTtccttttgactcttctgaactca | Andrew Basset lab; Welcome Sanger Institute | |
| LRRK2 Sanger sequencing F-AGGGACAAAGTGAGCACAGAA, | Andrew Basset lab; Welcome Sanger Institute | |
| LRRK2 Sanger sequencing R-CACAATGTGATGCTTGCATTTTT | Andrew Basset lab; Welcome Sanger Institute | |
| CRISPR guides and PCR primers for individual gene targets | This study | Table EV2 |
| **Chemicals, enzymes and other reagents** | | |
| Neomycin | Thermo Fisher Scientific | #21810031 |
| P3 buffer | Lonza | #V4XP-3032 |
| Accutase | Merck | #A6964 |
| SynthemaxII-SC-substrate | Corning | #3535 |
| Matrigel | Corning | #11543550 |
| RIPA | ThermoScientific | #89901 |
| AlphaLISA SureFire Ultra Human and Mouse Total LRRK2 Detection Kit | Revvity | #ALSU-TLRRK2-A-HV |
| Pierce BCA Protein Assay Kit | Thermo Fisher Scientific | #23227 |
| Geltrex LDEV-Free, hESC-Qualified | Thermo Fisher Scientific | #A1413302 |
| Essential 8 medium | Thermo Fisher Scientific | #A1517001 |
| Dulbecco's phosphate-buffered saline (DPBS) with calcium and magnesium | Thermo Fisher Scientific | # CM-PBS, 14040133 |
| SB 431542 | Tocris | #1614 |
| Dorsomorphin dihydrochloride | Thermo Fisher Scientific | #3093 |
| MEM Non-Essential Amino Acids | Thermo Fisher Scientific | #11140035 |
| Sodium pyruvate solution | Thermo Fisher Scientific | #S8636 |
| 2-Mercaptoethanol | Thermo Fisher Scientific | #31350010 |
| Insulin solution human | Thermo Fisher Scientific | #I9278 |
| N-2 Supplement (100X) | Thermo Fisher Scientific | #17502048 |
| DMEM/F-12 GlutaMAX Supplement | Thermo Fisher Scientific | #31331093 |
| Penicillin-Streptomycin (10,000 U/mL) | Thermo Fisher Scientific | #15140122 |
| GlutaMAX Supplement | Thermo Fisher Scientific | #35050038 |
| B-27 Supplement (50X), serum-free | Thermo Fisher Scientific | #17504044 |
| Neurobasal Medium | Thermo Fisher Scientific | #21103049 |

| Reagent/resource | Reference or source | Identifier or catalogue number |
|---|---|---|
| Dispase II | Thermo Fisher Scientific | #17105041 |
| Human FGF | Thermo Fisher Scientific | #146AA |
| Dimethyl sulfoxide (DMSO) | Thermo Fisher Scientific | # J66650.AE |
| RNA extraction kit | QIAGEN | #74104 |
| Nanostring panel | Strano A, Tuck E, Stubbs VE, Livesey FJ. Variable Outcomes in Neural Differentiation of Human PSCs Arise from Intrinsic Differences in Developmental Signalling Pathways. Cell Rep. 2020 Jun 9;31(10):107732. doi: 10.1016/j.celrep.2020.107732. PMID: 32521257; PMCID: PMC7296348. | |
| 4,6-diamidino-2-phenylindole (DAPI) | Thermo Fisher Scientific | #D1306 |
| Lipofectamine 3000 | Thermo Fisher Scientific | # L3000001 |
| Kanamycin | Thermo Fisher Scientific | #11588676 |
| IPTG | Thermo Fisher Scientific | #30018359 |
| Imidazole | Thermo Fisher Scientific | # 210330010 |
| Protease inhibitor cocktail | Merck | #11697498001 |
| HisTrapHP column | GE Healthcare, Cytiva | #17524801 |
| Superdex 200 16/60 gel filtration column | GE Healthcare, Cytiva | #28989335 |
| Spin concentrator | Millipore, Amicon | #UFC9030 |
| Heparin | Sigma, Merck | #2106 |
| 3-(N-morpholino)propanesulfonic acid (MOPS) | Sigma, Merck | #M1254 |
| Sarkosyl | Sigma, Merck | #61739 |
| Dylight 488 NHS ester | Thermo Fisher Scientific | #46402 |
| pHrodo | Thermo Fisher Scientific | # P36600 |
| N,N-Dimethylformamide | Sigma, Merck | # 227056 |
| tris[2-carboxyethyl]phosphine, Hydrochloride | Thermo Fisher Scientific | # T2556 |
| Superdex 200 Increase 10/300 GL | GE Healthcare, Cytiva | # 28990944 |
| Transferrin from Human Serum, Alexa Fluor™ 633 Conjugate | Thermo Fisher Scientific | # T23362 |
| Papain | Worthington biochemical Corporation | # LS003124 |
| Paraformaldehyde Solution, 4% in PBS | Thermo Fisher Scientific | # J19943.K2 |
| Bovine serum albumin | Sigma, Merck | # A7906/ A3983 |
| EDTA (0.5 M), pH 8.0 | Thermo Fisher Scientific | # AM9260G |
| QuickExtract DNA Extraction Solution | Cambio | # QE09050 |
| Agencourt AMPure XP | Beckman Coulter UK | # A63880 |
| β-amyloid (1–42) HiLyte™ Fluor 488 labelled | Anaspec | # AS-60479-01 |
| α-synuclein (1–140), HiLyte™ Fluor 488 labelled | Anaspec | # AS-55457 |
| Epidermal Growth Factor, Fluorescein Conjugate | Thermo Fisher Scientific | # E3478 |
| Dextran, Fluorescein, 70,000 MW, | Thermo Fisher Scientific | # D1823 |
| T7 endonuclease I | New England Biolabs | # M0689 |
| Human LRP1 Cluster IV Fc Chimera Protein | R&D systems | #5395-L4-050 |
| Human LRPAP Protein | R&D systems | #4296-LR-050 |
| | Tocris Bioscience | |
| Dynasore | Tocris Bioscience | # 2897 |
| PIK93 | Tocris Bioscience | # 6440 |
| Wortmannin | Tocris Bioscience | # 1232 |
| MLi-2 | Tocris Bioscience | |

| Reagent/resource | Reference or source | Identifier or catalogue number |
|---|---|---|
| GSK3357679A | GSK (this study) | |
| Brefeldin | Tocris Bioscience | # 1231 |
| Bafilomycin | Tocris Bioscience | # 1334 |
| Cytotoxicity Detection Kit (LDH) | Sigma, Merck (Roche) | # 11644793001 |
| EZlink sulfo-NHS-SS-biotin | Thermo Fisher Scientific | # 21331 |
| 2-Cyano-4-methylpyridine | Thermo Fisher Scientific | # L17148.06 |
| Triton X-100 | Sigma, Merck | # X100 |
| NuPAGE™ LDS Sample Buffer | Thermo Fisher Scientific | # NP0007 |
| NuPAGE™ Bis-Tris Mini Protein Gels, 4–12% | Thermo Fisher Scientific | # NP0323BOX |
| **Software** | | |
| WGE tool | https://wge.stemcell.sanger.ac.uk/ | |
| Fiji software | https://imagej.net/software/fiji/downloads | |
| FlowJo software | Becton, Dickinson & Company | |
| MAGeCK (0.5.8) | https://sourceforge.net/p/mageck/wiki/Home/ | |
| RStudio (Version 1.3.1093) | https://posit.co/download/rstudio-desktop/ | |
| igraph | https://datastorm-open.github.io/visNetwork/igraph.html | |
| visNetwork | https://datastorm-open.github.io/visNetwork/ | |
| Harmony analysis software | Perkin Elmer Life Sciences | |
| Prism Software | GraphPad | |
| **Other** | | |
| Lonza 4D nucleofector | Lonza | #AAF-1003X |
| Illumina MiSeq | Illumina | |
| Nanostring nCounter platform | | |
| Olympus Inverted FV1000 | Olympus | |
| SH800S Cell Sorter, Sony | Sony | |
| Avestin Emulsiflex C5 | | |
| Hielscher UP200St ultrasonicator | | |
| Sysmex/Partec CellTrics 50 µm filter | Wolf Labs | # 04-0042-2317 |
| BD LSRFortessa Cell analyser | | |
| BD Influx™ Cell Sorter | | |
| HiSeq2500 Rapid Run | Illumina | |
| Attune CytPix | Thermo Fisher Scientific | |
| CellCarrier96 | Perkin Elmer Life Sciences | # 6055308 |
| Opera-Phenix | Perkin Elmer Life Sciences | |
| Nunc 24-Well Plate, Round | Nunc | # 144530 |
| 0.5 mL Soft tissue homogenising CK14 tubes (Bertin) | Fisher Scientific | # 11588545 |
| SpeedBeads Magnetic Neutravidin-Coated particles | Sigma, Merck | # GE78152104010150 |
| ChemiDoc Imaging System | Bio-Rad | |

## Production and characterisation of human iPSC-derived cerebral cortex neurons

Human pluripotent stem cell lines used in this study were KOLF2-C1 (WTSIi018-B-1) and derivative constitutively expressing Cas9 protein (KOLF2-C1 Cas9). KOLF2-C1 Cas9 was generated by integration of a Cas9 transgene driven by a CAG promoter at the AAVS1 locus. Cells were nucleofected (Lonza) with recombinant enhanced specificity Cas9 protein (eSpCas9_1.1) (Slaymaker et al, 2016), a synthetic crRNA/tracrRNA (target site gggggccactaggga-caggattgg) and a homology directed repair template (https://www.addgene.org/86698/) followed by selection in neomycin (50 ug/ml; Thermo Fisher Scientific, #21810031) and clonal isolation (Bruntraeger et al, 2019). Integrated clones were identified using PCR across the homology arms and validated by Sanger sequencing of the entire transgene and measurements of Cas9

activity. Random integration was screened for using PCR within the antibiotic resistance gene of the template plasmid.

## Generation of isogenic gain and loss of function LRRK2 mutation iPSCs

Optimal gRNA design was performed using the WGE tool (https://wge.stemcell.sanger.ac.uk/) to minimise off-target effects. Full-length chemically modified sgRNA was purchased (Synthego) with the LRRK2-targeting gRNA sequence: ATTGCAAAGATTGCT-GACTA. CRISPR/Cas9 genome editing was performed as previously described (Bruntraeger et al, 2019). Briefly, eSpCas9_1.1/sgRNA ribonucleoprotein complexes and ssODNs were transfected into iPSCs by electroporation using nucleofection according to the manufacturer's instructions (Lonza 4D nucleofector, CA137, P3 buffer). To generate an isogenic KOLF2-C1 LRRK2 G2019S gain-of-function iPSC line, 500 pmol of a homology-directed repair template (sequence: TTTCACACTGTATCCCAATGCTGCCAT-CATTGCAAAGATTGCTGACTACAGCATTGCTCAGTACTGC TGTAGAATGGGGATAAAAACATCAGAGGGCACAC) was added to the nucleofection. Cells were allowed to grow to 75% confluence, dissociated using Accutase (Merck Sigma, #A6964) and plated onto a SynthemaxII-SC-coated 10 cm2 dish (Corning, #3535) at a low density. Individual colonies were picked manually into 96-well plates coated with Matrigel (Corning, #11543550) for expansion, and LRRK2 KO clones were identified by high-throughput sequencing. Sequencing was performed from PCR products using primers (F-ACACTCTTTCCCTACACGACGCTCTTCCGATCT-cagatacctccactcagcca, R-TCGGCATTCCTGCTGAACCGCTCTTCCGA TCTtcctttgactcttctgaactca), indexed and sequenced on an Illumina MiSeq instrument, and final clones validated by Sanger sequencing using primers (F-AGGGACAAAGTGAGCACAGAA, R-CACAATGTGATGCTTGC ATTTTT).

LRRK2 protein levels from RIPA (Thermo Scientific, #89901) extracted iPSC lysate were assessed using the AlphaLISA SureFire Ultra Human and Mouse Total LRRK2 Detection Kit (Revvity, #ALSU-TLRRK2-A-HV), excitation emission measurements (615 nm) were corrected with reactions without donor antibody mix and normalised to total protein concentration determined by the Pierce BCA Protein Assay Kit (#23227).

Directed differentiation of human iPSCs to cerebral cortex was carried out as described, with minor modifications (Shi et al, 2012a, 2012b). Briefly, iPSC grown on Geltrex LDEV-Free, hESC-Qualified (Cat. no A1413302) and maintained in Essential 8 medium (Cat. no A1517001) were differentiated using a chemical morphogen induced neural differentiation protocol, SMAD inhibition and retinoid signalling (1 mM SB 431542; Tocris #1614 and 1 μM Dorsomorphin dihydrochloride, Cat. no 3093) in neural maintenance medium (MEM non-essential amino acids, Cat. no 11140035 [5 mL], Sodium pyruvate solution Cat. no S8636 [5 mL], 2-mercaptoethanol Cat. no 31350010 [1 mL], Insulin solution human Cat. no I9278 [250 μL], N-2 Supplement (100X) Cat. no 17502048 [5 mL], DMEM/F-12 GlutaMAX Supplement Cat. no 31331093 [500 mL], Penicillin-Streptomycin (10,000 U/mL) Cat. no 15140122 [5 mL], GlutaMAX Supplement Cat. no 35050038 [5 mL], B-27 Supplement (50X) serum-free Cat. no 17504044 [10 mL], Neurobasal Medium #21103049 [500 mL]) for 12 days. Resulting in a neuroepithelial monolayer, which was dissociated

using Dispase II (Cat. no 17105041) and gently triturated to form ~1–2 mm cell aggregates these were reattached to Geltrex coated plated and maintained in neural maintenance media supplemented with 20 ng/mL Human FGF (#146AA) for 4 days. 17 days after neural induction, additional gentle cell aggregate disassociation steps were performed using Dispase to remove unwanted cell types, over the course of 8 days this was repeated two to three times. Finally, ~26 days after neural induction, cell cultures were dissociated using Accutase (Merck Sigma, #A6964) and either stored at −196 °C in 10% v/v DMSO or maintained in neural maintenance medium for downstream assays. All steps were performed at 37 °C and 5% $CO_2$ using Thermo Fisher Scientific reagents unless otherwise stated. To establish cortical identity and quality of neuronal induction, gene expression profiling was performed on a custom gene expression panel (Strano et al, 2020). RNA was isolated from iPSC cortical inductions 35 days after induction, using an RNA extraction kit (QIAGEN, #74104). Expression of genes in neurons (MAP2, MAPT and NGN2), cerebral cortex progenitor cells (EMX2, PAX6 and FOXG1), ventral telencephalon (NKX2-1 and LHX8), and mid-/hindbrain (HOXA and HOXB) was assessed in all neuronal inductions on the Nanostring nCounter platform. Immunofluorescent staining was performed on neurons as previously described (Moore et al, 2015), using the following primary antibodies: tau (Dako, A0024), MAP2 (abcam; ab5392), Tuj1 (Cambridge Bioscience; ab14545), TBR1 (Abcam; ab31940), CTIP2 (Abcam; ab18465) EEA1 (Abcam; ab109110) and LAMP1 (Abcam; ab62562). Secondary antibodies used were as follows: anti-mouse Alexa 647 (A32787), anti-rat Alexa 647 (A48265), goat anti-chicken Alexa 488 (A11039), goat anti-rabbit Alexa 546 (A11010, all from Thermo Fisher Scientific). Samples were stained with DAPI (1:5000 in PBS). Images were acquired through an Olympus Inverted FV1000 confocal microscope and processed using the Fiji software (Schindelin et al, 2012).

## Cas9 editing efficiency in human iPSC neurons

Knockout efficiency in KOLF2-C1 Cas9 neurons was assessed using a lentiviral reporter system (Tzelepis et al, 2016). Briefly, neurons were transduced (30 days after induction) with lentiviruses expressing fluorophores mCherry and GFP and an sgRNA targeting GFP (pKLV2-U6gRNA(gGFP)-PGKmCherry2AGFP-W; modified from Tzelepis et al, 2016. Neurons were dissociated (Accutase [Merck Sigma, #A6964]) 5, 12, 17 and 29 days after transduction and analysed by flow cytometry (SH800S Cell Sorter, Sony). Knockout efficiency was calculated as the ratio of mCherry positive, GFP-negative neurons over total transduced cells (mCherry positive, GFP positive and negative) (FlowJo software). Data are shown from one experiment; similar results were obtained during assay optimisation.

## Genome-wide CRISPR RNA guide lentiviral library

The Human CRISPR Library v.1.1 was used in this study. The Human CRISPR Libraries v.1.0 and v1.1 have been previously described (Addgene, 67989) (Bruntraeger et al, 2019; Behan et al, 2019). CRISPR Library v.1.1 contains all the guides from v1.0 targeting 18,009 genes with 90,709 sgRNAs, plus five additional sgRNAs against 1876 selected genes (Wang et al, 2014). Lentiviruses were produced by transfecting HEK 293 cells with

lentiviral vectors together with packaging vectors (third generation) using Lipofectamine 3000 (Thermo Fisher). Viral supernatant was collected 24 and 48 h post-transfection, filtered, concentrated by centrifugation and stored at −80 °C. The library was sequenced to assess the distribution of the guides. Virus MOI was determined by flow cytometry (SH800S Cell Sorter, Sony; FlowJo software). For each whole-genome CRISPR knockout screen, $4-6 \times 10^7$ neurons (35 days after induction) were transduced with an appropriate volume of the lentiviral packaged whole-genome sgRNA library to achieve 30-40% transduction efficiency (>100×library coverage).

## Recombinant tau monomer and aggregate preparation

Recombinant tau protein was purified as previously described (Evans et al, 2018). Briefly, tau P301S_FLAG×3_10×his-tag was overexpressed using the pET24d plasmid in BL21(DE3) bacteria (37 °C, 50 mg.mL$^1$ Kanamycin, IPTG 0.4 mM; both Fisher Scientific, #11588676 and #30018359). Cells were lysed (Avestin Emulsiflex C5) in lysis/binding buffer (20 mM phosphate [pH7.4], 500 mM NaCl, 20 mM imidazole, protease inhibitor cocktail [Merck #11697498001]), clarified lysate was applied to a 5 mL HisTrapHP column (GE Healthcare, Cytiva #17524801) and washed with 10 CV of lysis/binding buffer. Tau was eluted in 20 mM phosphate (pH 7.4), 500 mM NaCl, 500 mM imidazole. Peak fractions were pooled and further purified using a Superdex 200 16/60 gel filtration column (GE Healthcare, Cytiva #28989335) in 50 mM phosphate (pH 7.4), 150 mM NaCl. Pooled fractions were then concentrated to ~8 mg/mL using a spin concentrator (Millipore, Amicon #UFC9030). Tau fibrils were prepared following the protocol and characterisation as described (Evans et al, 2018). About 1 mL tau P301S at 8 mg/mL was incubated with 4 mg.mL$^{-1}$ heparin (Sigma) in PBS plus 30 mM 3-(N-morpholino) propanesulfonic acid (MOPS) (pH 7.2) at 37 °C for 72 h. Fibrillar material was diluted in 9 mL PBS plus 1% (v/v) sarkosyl (Sigma, Merck #61739) and left rocking for 1 h at room temperature to completely solubilise any non-fibrillar material. Insoluble tau was pelleted by ultracentrifugation for 1 h at 4 °C. The pellet was resuspended in 1 mL PBS by vigorous pipetting and sonicated at 100 W for $3 \times 20$ s (Hielscher UP200St ultrasonicator) to disperse clumps of protein and break large filaments into smaller species. Final protein concentration was determined by the Pierce BCA Protein Assay Kit (23227).

To label purified recombinant tau (monomeric or fibrillar tau P301S) with Dylight 488 NHS ester (Thermo Fisher Scientific #46402) or pH-sensitive form of rhodamine (pHrodo, Thermo Fisher Scientific # P36600), 150 µM tau protein (or equivalent protein concentration for aggregate; ~7 µg.mL$^{-1}$) was incubated with either a tenfold-molar excess of Dylight(10 mg.mL$^{-1}$; dissolved in DMF) or pHrodo (10 mM; dissolved in DMSO; with tenfold-molar excess of tris[2-carboxyethyl]phosphine; TCEP) for 2 h in the dark at room temperature. After incubation, labelled protein samples were subjected to size exclusion chromatography at 4 °C (Superdex 200 Increase 10/300 GL, GE Healthcare) in 50 mM phosphate (pH 7.4) and 150 mM NaCl to remove unreacted dye and assess perturbation of oligomeric state by labelling (our previous studies show no effect upon labelling (Evans et al, 2018). About 64.5 µM Transferrin from Human Serum, Alexa Fluor™ 633 Conjugate (T23362; Thermo Fisher Scientific) was dissolved in PBS.

## Flow cytometry-based whole-genome screen for genes modifying tau entry and processing

Neurons transduced with the Human CRISPR library were matured in neuronal maintenance medium (N2B27; media changed every two days) for 65 days. Twenty-four hours prior to acute incubation with labelled recombinant proteins, media was exchanged to DMEM plus supplements (500 ml DMEM/F-12 + GlutaMAX, 0.25 mL Insulin (4 mg.mL$^{-1}$), 1 mL 2-mercaptoethanol (50 mM), 5 mL MEM non-essential amino acids Solution, 5 mL sodium pyruvate (100 mM) and 2.5 mL Penicillin-Streptomycin (10,000 U/mL); Thermo Fisher; note this is without N$_2$ supplement that contains transferrin) to remove transferrin from the culture medium. The following day, neurons were incubated with 30 nM of transferrin-Alexa 633 and either 90 nM monomeric or 400 nM fibrillar tau-Dylight for 4 or 5 h, respectively. After two PBS washes, neuronal cultures were dissociated with papain (Worthington Biochemical Corporation) supplemented with Accutase (Merck Sigma, #A6964) into single cells.

Dissociated neurons were collected by low-speed centrifugation ($600 \times g$ for 15 min), the supernatant was removed, filtered (Sysmex/ Partec CellTrics 50 µm filter, Wolf Labs #04-0042-2317) and fixed using 2% paraformaldehyde in PBS for 15 min at room temperature. Neurons were washed twice in PBS, resuspended in PBS, 7.5% w/v BSA, 2 mM EDTA and analysed by flow cytometry. Flow cytometry assays were performed on a Sony SH800 Cell Sorter or BD LSRFortessa Cell. For large-scale CRISPR knockout screens, neurons were collected using a BD Influx cell sorter. Neurons were gated using forward and side scatter channels to isolate the viable cell population and exclude noise/debris, subsequently gated on forward scatter and trigger pulse width channel to isolate singlets. MAP2-positive cells were identified using Milli-Mark Anti-MAP2-PE antibody (Sigma-Aldrich; FCMAB318PE). Transferrin and tau uptake assays were gated for fluorescence in the channels corresponding to BFP (sgRNA library), Dylight 488 (tau) and Alexa 633 (transferrin), using non-transduced neurons and neurons without acute incubation of recombinant proteins to establish thresholds. Neuronal populations were collected that were positive for the sgRNA library and had high levels of labelled transferrin and either high or low levels of labelled tau protein.

## Guide RNA sequencing

Genomic DNA was extracted from cell pellets using QuickExtract DNA Extraction Solution (Cambio; QE09050) as per the manufacturer's instructions. sgRNA amplification was performed using a two-step PCR strategy and purified using Agencourt AMPure XP (A63880; Beckman Coulter UK), Illumina sequencing (19-bp single-end sequencing with custom primers on the HiSeq2500 Rapid Run) and sgRNA counting were performed as described previously (Tzelepis et al, 2016).

## CRISPR screen data analysis

sgRNA count files from purified neuronal populations that had high levels of labelled transferrin and tau were compared to neuronal populations with high levels of labelled transferrin and low levels of labelled tau protein using MAGeCK (0.5.8) (Li et al, 2014). Data from replicate monomeric ($n = 2$) or fibrillar ($n = 3$) tau

uptake screens were grouped in the analysis. sgRNA rankings were summarised into gene-level statistics using the RRA algorithm. All subsequent analyses were performed in RStudio (Version 1.3.1093). For genes with two sets of sgRNAs, the higher-ranking set was kept and the lower-ranking set was discarded. An empirical threshold of unadjusted $p$ value <0.01 was used to identify genes required for tau uptake in either screen, identifying 214 genes in the fibrillar screen and 228 genes in the monomeric screen. Functional term enrichment analysis was performed using gProfiler 2, querying the GO, REAC, and CORUM databases (Raudvere et al, 2019), for the term enrichment analysis, individual gene names from the library were manually updated to their most current name as indicated in Dataset EV1.

Compartment localisation enrichment was performed using localisation scores from the COMPARTMENTS database (Binder et al, 2014). Localisation of each gene to a particular compartment is given a score of up to 5 based on the strength of the supporting evidence. To calculate compartment localisation enrichment for the genes identified in the CRISPR screens, a gene-set localisation score for each compartment was calculated by averaging the localisation score of all constitutive genes. The score ranged from 0 to 1, with 1 representing the maximum localisation score for that compartment for all the genes in the set. Cellular compartments were removed from analysis if too undescriptive (gene-set localisation score ≥0.7), with low signal (gene-set localisation score ≤0.02), or with only weak evidence (no individual gene scoring 3.5/5 or above). Compartment localisation enrichment was calculated compared to the mean value obtained from 100,000 simulated random sets of 228 (fibrillar), 214 (monomeric), 431 (combined fibrillar and monomeric) or 21 (combined fibrillar and monomeric, and either Zika, Influenza A, or Coronavirus) genes from the CRISPR library. Boot-strapping was used to calculate $p$ values, and significance was adjusted for multiple comparison using the false discovery rate. For significantly enriched compartments, the log2 fold change of compartment score over random set average was mapped onto a diagram of cellular compartments using the sp package (Renard, 2011).

Protein–protein interaction networks were constructed using the PSICQUIC package using the following databases: BioGrid, bhf-ucl, IntAct, MINT, UniProt, MBInfo, InnateDB (Aranda et al, 2011). All PPIs for the human proteins were maintained, removing reciprocal and self-interactions. Network graphs were drawn using the igraph and visNetwork packages (Almende et al).

## Flow cytometry analysis of extracellular protein uptake in neurons with individual gene perturbations

KOLF2-C1 Cas9 neurons were transduced with lentiviruses (30 days after induction) containing vectors expressing CRISPR-Cas9 guides targeting individual genes (Dataset EV5), pKLV2-U6gRNA5(BbsI)-ccdb-PGKpuro2ABFP-W (modified from Addgene, 67974). Transduced or CRISPR-engineered (LRRK2 gain and loss of function mutations) neurons were maintained (65 days) and subsequently assayed for their ability to take up recombinant extracellular protein as previously described, with modifications to protein concentrations and incubation times. Neurons were incubated with 90 nM of transferrin-Alexa 633 and either 90 nM monomeric (wild-type or P301S; 4 h incubation), 400 nM fibrillar tau-Dylight 488 (5 h incubation), 250 nM β-amyloid (1–42)

HiLyte™ Fluor 488 (5 h incubation; Anaspec; AS-60479-01), 100 nM α-synuclein (1–140) HiLyte™ Fluor 488 labelled (5 h incubation; Anaspec; AS-55457), 150 nM epidermal growth factor, fluorescein conjugate (3 h incubation; Thermo Fisher Scientific; E3478), or 500 nM Dextran, Fluorescein, 70,000 MW, (3 h incubation; Thermo Fisher Scientific; D1823). Dissociated neurons were analysed by flow cytometry (BD Fortessa Cell Analyser). For LRRK2 G2019S neurons, uptake of tau was measured in independent experiments at two sites (Sanger Institute and Babraham Institute, Cambridge), using two different FACS platforms (BD Fortessa and Thermo Fisher Attune CytPix). Gated singlet neurons were assessed for fluorescence in the channels corresponding to the labelled proteins above, and thresholds were established as described previously. Flow cytometry data were analysed using FlowJo software. To confirm individual CRISPR guide targeting, T7 endonuclease I assays were performed on a subset of targeted genes. Genomic DNA was isolated from KOLF2-C1 Cas9 neurons (~65 DIV) transduced with lentivirus containing guide RNA targeting a gene of interest or GFP (non-targeting control). Genomic regions containing target sites for CRISPR guides were amplified by PCR using primer pairs. Purified PCR products were incubated with T7 endonuclease I (New England Biolabs; #M0689) for 15 min at 37 °C and analysed by agarose gel (ethidium bromide).

## Live imaging of tau entry to human iPSC neurons

Neurons grown (65 days after induction) on 96-well plates (CellCarrier96; Perkin Elmer Life Sciences; 6055308) were imaged for fluorescent pHrodo-labelled tau protein (excitation at 577 nm and emission at 641 nm in an acidic environment) using an Opera-Phenix (Perkin Elmer). Bright-field and fluorescence emission images were collected at 45 min or 1 h intervals for between 4 and 10 h at 37 °C, 5% $CO_2$. Parameters for fluorescent objects were set (fluorescent intensity and contrast), quantified and the intensity of the field was normalised to confluency using Harmony Analysis Software. Data were analysed using Prism Software (GraphPad). Typically, conditions were repeated at least in triplicate and nine fields were recorded per well.

Live imaging assays were performed using monomeric (50 nM) or fibrillar (100 nM, equivalent calculated from monomeric tau; 5 μg.mL$^{-1}$) pHrodo-labelled tau. For competitive binding experiments, recombinant peptides were resuspended in PBS, and PBS was the vehicle control in experiments. Recombinant proteins were separately pre-incubated with neuronal cultures and recombinant tau protein for 3 h prior to incubation of neuronal cultures with the mixer containing recombinant proteins (100 nM Human LRP1 Cluster IV Fc Chimera Protein #5395-L4-050; 10 nM Human LRPAP Protein #4296-LR-050 [R&D Systems]). For pharmacological treatments, compounds were dissolved in DMSO at the concentrations noted, and DMSO was the vehicle control in experiments. A 1-week pharmacological treatment regime of neurons, exchanging media every 48 h, was employed for Dynasore (2897), PIK93 (6440), wortmannin (1232), MLi-2 (5756; Tocris Bioscience), GSK3357679A (GSK) or vehicle control (0.1% [v/v] DMSO). A 24 h treatment regime of neurons was employed for Brefeldin (1231), Bafilomycin (1334; Tocris Bioscience) or vehicle control (0.1% [v/v] DMSO). Neurotoxicity/neurolysis were assessed by analysing the level of lactate dehydrogenase (LDH) activity

(Roche; 11644739001) in conditioned media samples following treatments.

## Determination of relative surface protein levels

Neurons were grown (65 days after induction) on 24-well plates (Nunc 24-Well Plate, Round, 144530). Following steps were carried out on ice. Neurons were washed three times with 0.5 mL Dulbecco's phosphate-buffered saline (DPBS) with calcium and magnesium (CM-PBS, 14040133; Thermo Fisher Scientific). Neurons were then incubated with biotin buffer (0.2 mg.ml$^{-1}$ EZlink sulfo-NHS-SS-biotin, 21331, Thermo Fisher Scientific in CM-PBS) for 30 min, followed by replacement of Biotin buffer with quenching reagent (50 mM NH4Cl in CM-PBS) and left for 10 min. Following a repeat of the quenching step and three subsequent washes in CM-PBS, neurons were resuspended in RIPA buffer (Thermo) containing cOmplete Protease Inhibitor Cocktail (Roche; 11697498001) and transferred to 0.5 mL soft tissue homogenising CK14 tubes (Bertin) and lysed using the Precellys Evolution system. Neuronal lysates were left for 5 min and centrifuged at 16,000×$g$ for 10 min. The protein concentration of isolated lysate supernatant was measured using the Pierce BCA Protein Assay Kit (23227). Normalised protein lysates (cell lysate) were incubated overnight with 40 µL (50% slurry) of SpeedBeads Magnetic Neutravidin-Coated particles (GE781152104010150) on a rotator. Surface biotinylated proteins bound to Neutravidin beads were separated from unlabelled intracellular proteins by magnetism, washed four times in 1 mL of wash buffer (10 mM Tris (pH 7.4), 1.5 mM EDTA, 150 mM NaCl, 1% Triton X-100, containing cOmplete Protease Inhibitor Cocktail). After centrifugation at 1500×$g$ for 30 s and removal of residual wash buffer, surface proteins bound to beads were resuspended in 50 uL of NuPAGE™ LDS sample buffer (NP0007) and denatured at 95 °C for 10 min. Resulting samples (cell lysate and surface fractions) were analysed by western blot using antibodies for LRP1 (Abcam; ab92544; 1:1000), N-Cadherin (ab18203) and β-Actin (Sigma-Aldrich; A2228; 1:1000). Primary antibodies were used at 1:1000 dilution and incubated overnight in PBS with 3% (w/v) Bovine Serum Albumin (Sigma-Aldrich; A3983), and detected using secondary IgG antibodies conjugated to fluorescent probes (Alexa Fluor 488; A-2120; Alexa Fluor 546; A10040). Protein levels were assessed using a ChemiDoc Imaging System and the Image Lab software (Bio-Rad).

## Statistical analysis

The number of replica wells and experiments are indicated in the figure legends for each assay where appropriate. For live-cell imaging assays using pHrodo-labelled protein, intensity was normalised to the area of the field occupied by cells using the Harmony Analysis Software (Revvity). Typically, nine fields were recorded per well, and an average was generated from the Harmony Analysis Software. Analyses were performed using the Prism version 8 software (GraphPad). One-way ANOVA with Dunnett's multiple testing or Student's $t$-test was used where appropriate.

Hypergeometric tests were used to evaluate the significance of overlap between gene sets. Genes required for uptake of Ebola, influenza A, Zika and SARS-CoV-2 viruses, and for different size beads were obtained from published CRISPR screens [35, 36, 38] and were defined as having an FDR <0.05. Genes required for uptake of transferrin and EGF were obtained from a published RNAi screen (Collinet et al, 2010) and defined as having χ2 probability $(1.0 - P) \geq 0.95$. Hypergeometric tests for overlap in screen hits were carried out using the set of genes analysed in both studies as background.

## Data availability

The source data of this paper are collected in the following database record: https://www.ebi.ac.uk/biostudies/studies/S-BSST2010.

## Peer review information

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

## Acknowledgements

The authors thank Sarah Spain, Paris Litterick, Alison Mann, Gosia Trynka, David Hulcoop and Ian Dunham of the Open Targets team at the European Bioinformatics Institute and Wellcome Sanger Institute for ongoing support of this research; Fiona Behan, Chun-Fang Xu (GSK) and Jinkuk Choi (Biogen) for feedback on data interpretation; Sam Thompson, Jennie Graham, Chris Hall and Bee Ling Ng for flow cytometry support, Kathleen Long and Michaela Bruntraeger who generated iPSC cell lines (Sanger Institute); Tom Campbell, Sergey Sitnikov and Clare Jones for technical assistance (Talisman Therapeutics) and Anne Hedegaard (University of Oxford) for providing reagents. This work was funded by Open Targets (OTAR036, www.opentargets.org). This research was also supported by Wellcome (WT101052MA Investigator Award to FJL), Alzheimer's Research UK (Stem Cell Research Centre), Dementias Platform UK (Stem Cell Network) and Great Ormond Street Hospital Charity (Stem Cell Professorship). This research was supported in part by the NIHR Great Ormond Street Hospital Biomedical Research Centre. The views expressed are those of the author(s) and not necessarily those of the NHS, the NIHR, or the Department of Health. This research was funded in whole, or in part, by the Wellcome Trust 220540/Z/20/A. For the purpose of Open Access, the author has applied a CC BY public copyright licence to any Author Accepted Manuscript version arising from this submission.

## Author contributions

**Lewis D Evans**: Conceptualisation; Formal analysis; Investigation; Methodology; Writing—original draft; Writing—review and editing. **Alessio Strano**: Formal analysis; Writing—original draft. **Eleanor Tuck**: Investigation. **Ashley Campbell**: Investigation. **James Smith**: Investigation. **Christy Hung**: Investigation. **Tiana S Behr**: Investigation. **Bernardino Ghetti**: Investigation. **Benjamin Ryskeldi-Falcon**: Investigation. **Emre Karakoc**: Formal analysis. **Francesco Iorio**: Formal analysis. **Alastair Reith**: Funding acquisition; Project administration. **Andrew R Bassett**: Funding acquisition; Investigation; Methodology. **Frederick J Livesey**: Conceptualisation; Funding acquisition; Investigation; Writing—original draft; Project administration; Writing—review and editing.

Source data underlying the figure panels in this paper may have individual authorship assigned. Where available, figure panel/source data authorship is listed in the following database record: BioStudies: S-BSST2010.

## Disclosure and competing interests statement

Open Targets is a public–private partnership between non-profit research institutions and the pharmaceutical industry. FJL is a founder and holds equity in Gen2 Neuroscience and Talisman Therapeutics.

# Expanded View Figures

**Figure EV1.   Development and optimisation of CRISPR screens for tau uptake in human iPSC-derived excitatory neurons.**     ▶

(A) Construct of KOLF2-C1 Cas9 cell line. KOLF2-C1 parent cell line genetically engineered at the AAVS1 locus to constitutively express humanised Cas9 with N-terminal FLAG tag (three repeats) and SV40 nuclear localisation signal (NLS) and C-terminal nucleoplasmin NLS from CAG promoter (CMV/chicken β-actin promoter). (B) Gene expression analysis of iPSC-derived neuronal progenitor cells from KOLF2-C1 (parental line) and KOLF2-C1 Cas9 (constitutively expressing Cas9), 35 days after induction. Selected genes whose expression is specific or enriched in particular regions and/or cell types are shown from a panel of 200 genes (see Experimental Procedures for details). (C) Analysis of the proportion of MAP2-positive KOLF2-C1 Cas9 neurons 60 days after induction. Neurons were dissociated into single cells and immunostained using MAP2 antibody conjugated to PE and analysed by flow cytometry. To establish threshold levels for MAP2-positive neurons (bold horizontal bar), unstained control neurons were analysed. (D, E) Optimising the concentration of tau protein to achieve saturating levels of extracellular tau uptake during acute treatment, measured by flow cytometry. Graph reports percentage of tau-positive neurons over time after tau incubation initiation, with concentration of tau protein as indicated. Data are shown from one representative experiment. (F) Percentage of whole-genome CRISPR gRNA library lentivirus-transduced (BFP+) neurons in tau uptake FACS screens. A monomeric tau uptake screen was carried out in duplicate, and a fibrillar tau uptake screen in triplicate. (G) Total number of lentivirus-transduced (BFP+) neurons collected by flow cytometry in each screen (left axis) and corresponding library coverage (n. cells/n. gRNAs) (right axis). (H) Percentage of lentivirus-transduced neurons collected from monomeric and fibrillar tau uptake screens that are transferrin positive and tau negative. (I) Total number of transferrin-positive, tau-negative lentivirus-transduced neurons collected by flow cytometry in each screen (left axis) and corresponding library coverage (n. cells/n. gRNAs) (right axis). (J) Distribution of median normalised log2 guide RNA counts from transferrin positive and either tau positive (+) or negative (−) neurons collected from monomeric (purple; two screens) and fibrillar (orange; three screens) tau uptake screens. (K) Percentage of CRISPR gRNA detected in transferrin-positive and either tau-positive (POSPOS) or negative (NEGPOS) neurons collected from tau uptake FACS screens.

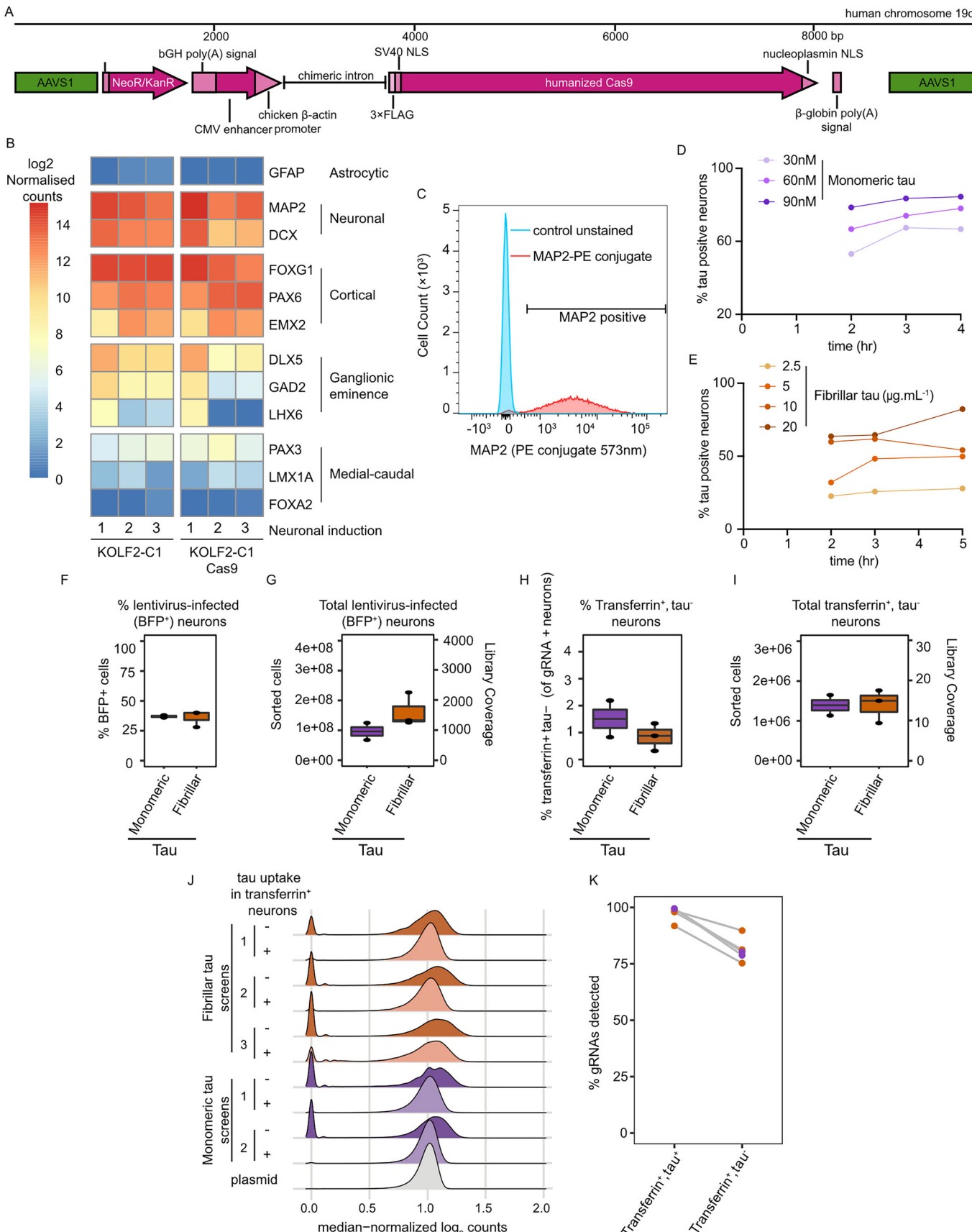

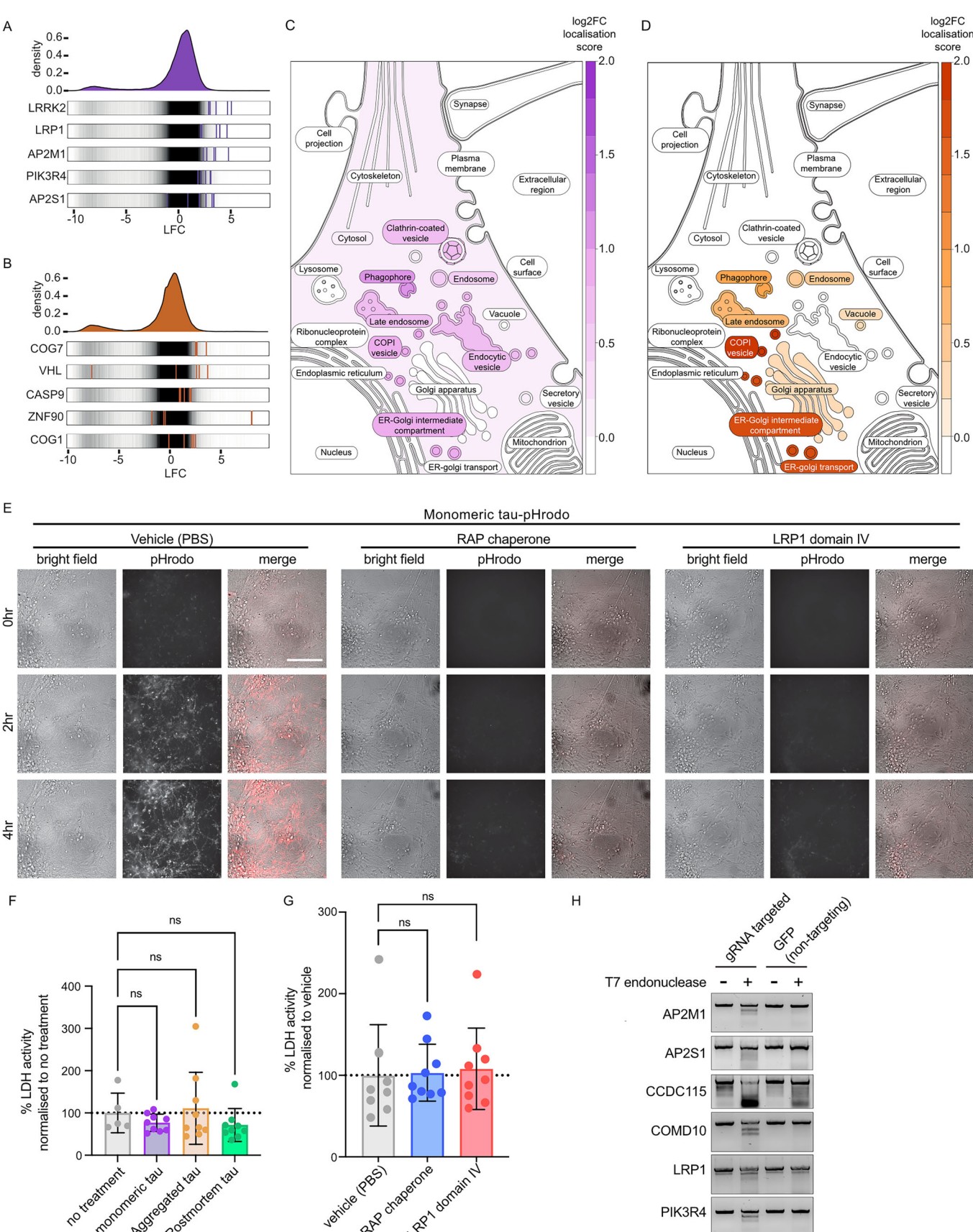

**Figure EV2. Identification of genes involved in the uptake of structurally distinct forms of tau by human cortical neurons via low-pH intracellular compartments.**

(A, B) CRISPR gRNA log fold change (LFC) between transferrin positive and either tau positive (+) or negative (−) neurons collected from monomeric (A) and fibrillar (B) tau uptake screens analysed using the MAGeCK algorithm. Guides for the five highest-ranked genes (gene names indicated) are highlighted on the guide density plots. Genes required for uptake of monomeric (C), fibrillar (D) tau code for proteins with a significantly higher than random localisation score in particular cellular compartments in the COMPARTMENTS dataset (FDR <0.05). Significantly enriched compartments are coloured based on the strength of enrichment (log2 fold change), whereas non-significant compartments are left white. (E) Time-lapse (0- to 4-h) images showing uptake into iPSC-derived human neurons (60 days after induction) of extracellular monomeric tau conjugated to a pH-sensitive dye (inverse relationship between fluorescence and pH). Neurons and tau protein were individually pre-incubated with either 10 nM RAP chaperone, 100 nM LRP1 domain IV peptide or vehicle control (PBS) for 3 h prior to combining the tau incubations with neurons and live imaging of neuronal uptake of tau. Bright-field (grey scale in merge) and pH-sensitive fluorescent signal (pHrodo; red in merge) were captured using automated imaging on the Opera-Phenix platform (Perkin Elmer). Scale bar, 100 μm. (F) None of the forms of tau were acutely toxic to neurons over a 16 hr period, as measured by extracellular LDH activity (three wells per treatment), in the presence of 25 nM Monomeric tau, 150 nM (monomer molar equivalent) fibrillar or post-mortem tau. (G) Extracellular LDH activity was also used to determine neuronal viability in the presence of vehicle (PBS), 10 nM RAP chaperone or 100 nM LRP1 domain IV peptide (after treatment for one week; >8 wells per treatment, across two biological replicates). Error bars indicate SD. Significance was determined using one-way ANOVA ($*p < 0.05$, $**p < 0.01$, Dunnett's test for multiple comparisons). (H) The T7 endonuclease assay was used to confirm CRISPR guide RNA targeting. Assays were performed on amplified genomic DNA regions containing the target site for CRISPR gRNAs to genes indicated in the presence of targeting gRNA and the non-targeting GFP control.

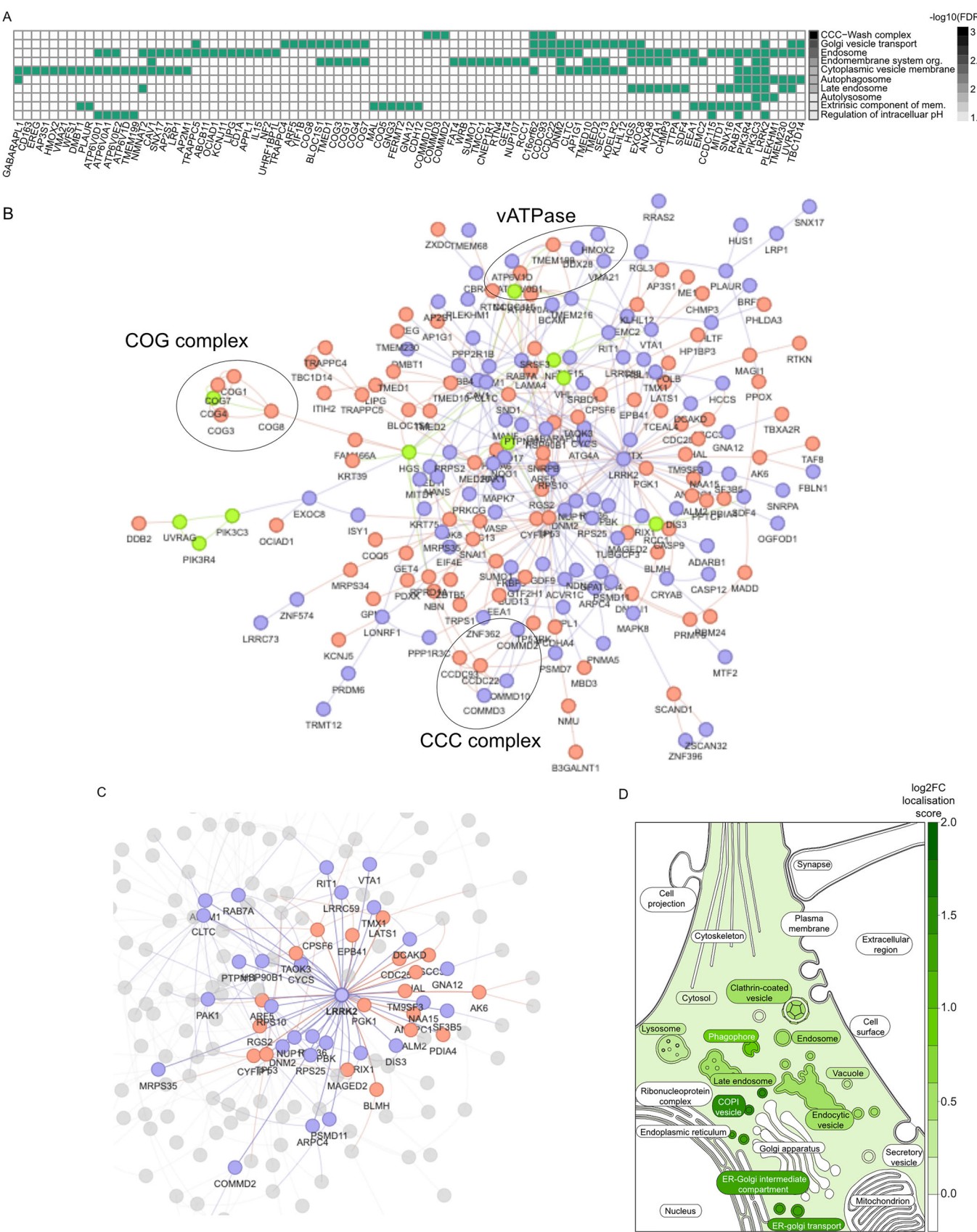

◀ **Figure EV3. Comparison of genes required for monomeric and fibrillar tau entry by human cortical neurons.**

(A) Heat map showing a representative selection of significantly enriched terms annotating the 431 genes significantly enriched for either monomeric or fibrillar tau uptake. The black colour scale indicates the significance level ($-$log10 FDR). Rows are sorted in order of significance, also see Dataset EV3. The central heatmap shows which of the enriched genes is annotated with each term. (B) PSICQUIC-derived network of experimentally validated physical interactions between proteins encoded by genes identified in either screen. Nodes are colour-coded depending on whether the corresponding gene was identified as required for monomeric (purple) or fibrillar (orange) tau uptake, or both (green). Some notable complexes are highlighted with circles. Interactions disconnected from the main network are not included. (C) The direct interactors of LRRK2, the most connected node, are highlighted in the protein interaction network; indirect interactions and genes unconnected to LRRK2 appear in grey. (D) Genes required for uptake of either monomeric or fibrillar tau code for proteins with a significantly higher than random localisation score in particular cellular compartments in the COMPARTMENTS dataset (FDR <0.05). Significantly enriched compartments are coloured based on the strength of enrichment (log2 fold change), whereas non-significant compartments are left white.

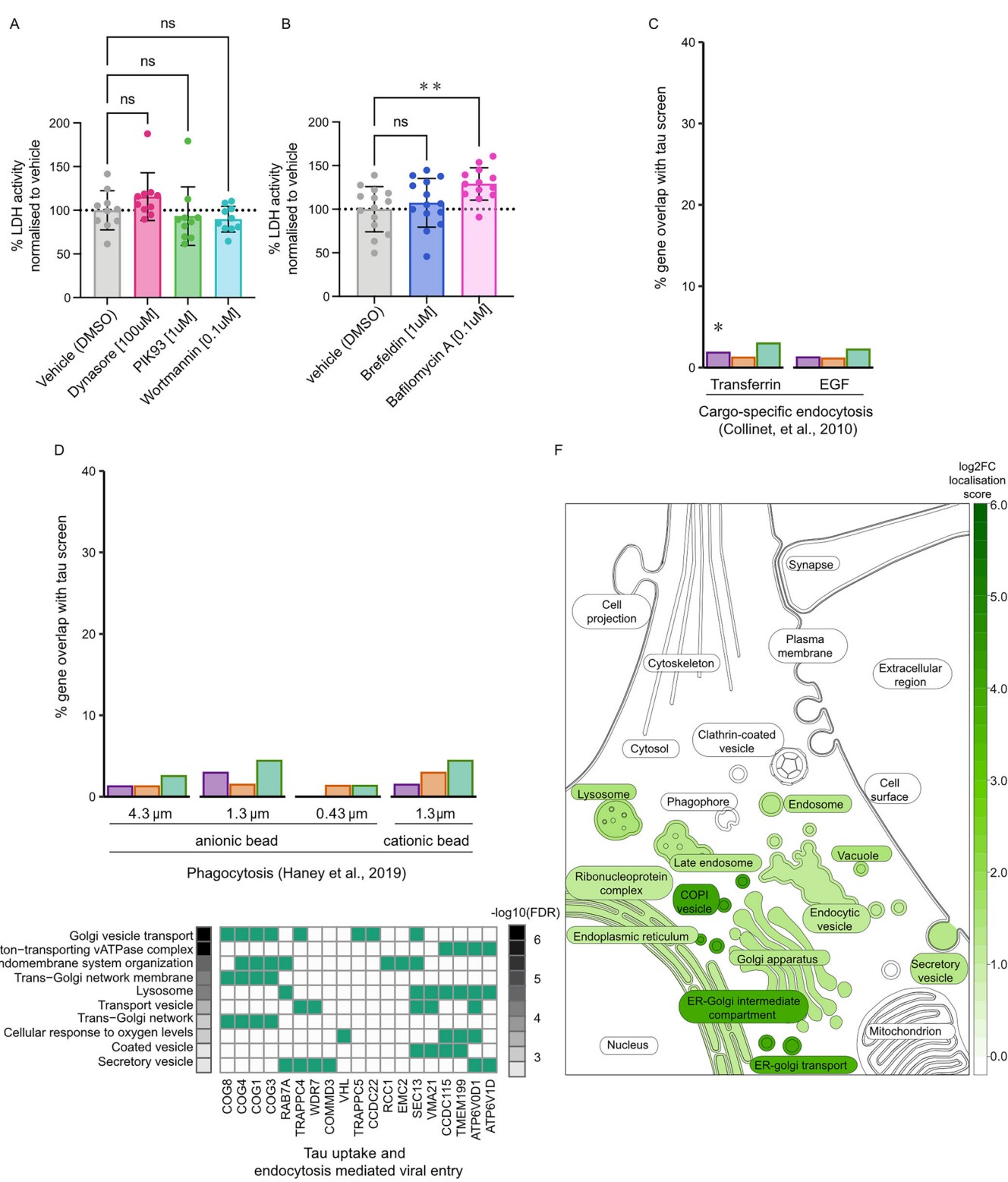

**Figure EV4. Cellular mechanisms for tau uptake and processing by human excitatory neurons in relation to endocytosis and viral entry, CRISPR screens.**

(A) Extracellular LDH activity was used to determine neuronal viability in the presence of vehicle (0.1% [v/v] DMSO), 100 μM Dynasore, 1 μM PIK93 or 0.1 μM Wortmannin (after treatment for one week) (B), and for 1 μM Brefeldin or 0.1 μM Bafilomycin A (after treatment for 24 hr) (I). Only Bafilomycin A had a modest effect on neuronal viability (ten wells per treatment). Error bars indicate SD. Significance was determined using one-way ANOVA (*$p < 0.05$, **$p < 0.01$, Dunnett's test for multiple comparisons). (C) Neuronal uptake of tau protein requires cargo-specific endocytic adaptors. Comparison of sets of genes required for tau uptake with genes identified in screens for endocytosis of either transferrin or epidermal growth factor (EGF), shows significant overlap between monomeric tau uptake and transferrin endocytosis, but no other significant overlaps between the gene sets. (D) Genes involved in tau uptake do not significantly overlap with sets of genes identified as phagocytosis regulators in screens performed with distinct substrates, varying in their diameter (μm) and charge. (E) Heatmap showing functional annotations enriched among genes required for the uptake of either form of tau and Influenza A, Zika or SARS-CoV-2 virus entry (representative term selection). (F) Genes required for either monomeric or fibrillar tau uptake and Influenza A, Zika or SARS-CoV-2 viral entry code for proteins with a significantly higher than random localisation score in particular cellular compartments in the COMPARTMENTS dataset (FDR <0.05). Significantly enriched compartments are coloured based on the strength of enrichment (log2 fold change), whereas non-significant compartments are left white.

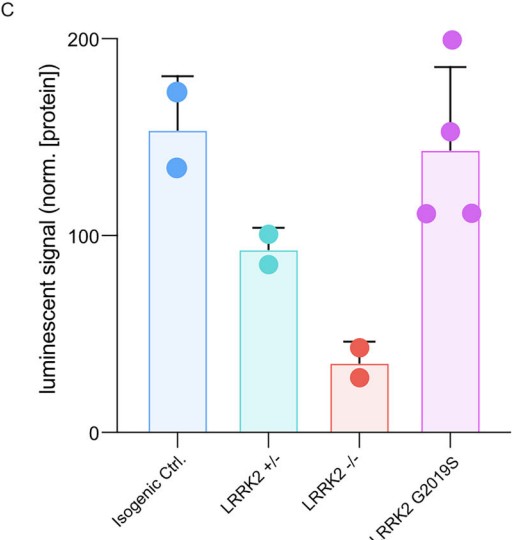

◀ **Figure EV5. Neuronal gene expression and LRRK2 protein levels in iPSC-derived cortical neurons with LRRK2 gain and loss of function mutations.**

(A, B) Gene expression (Nanostring) analysis of iPSC-derived neuronal progenitor cells from KOLF2-C1 (parental line; Isogenic control), LRRK2 heterozygous null (LRRK2 −/+), LRRK2 homozygous null (LRRK2 −/−) and LRRK2 heterozygous G2019S, 35 (A) and between 64–76 (B) days after induction. Selected genes whose expression is specific or enriched in particular regions and/or cell types are shown from a panel of 200 genes (see Experimental Procedures for details), demonstrating that the inductions generated primarily cortical progenitor cells, with some ventral interneuron progenitor cells, which subsequently generate excitatory and inhibitory neurons. (C) LRRK2 protein levels in iPSC from KOLF2-C1 (parental line; Isogenic control), LRRK2 heterozygous null (LRRK2+/−), LRRK2 homozygous null (LRRK2−/−) and LRRK2 heterozygous G2019S, measured by AlphaLISA assay (see Methods for details; at least two wells per genotype). LRRK2 +/− cells have reduced LRRK2 protein relative to the parental cell, whereas LRRK2 −/− cells have protein around the detection threshold of the assay.

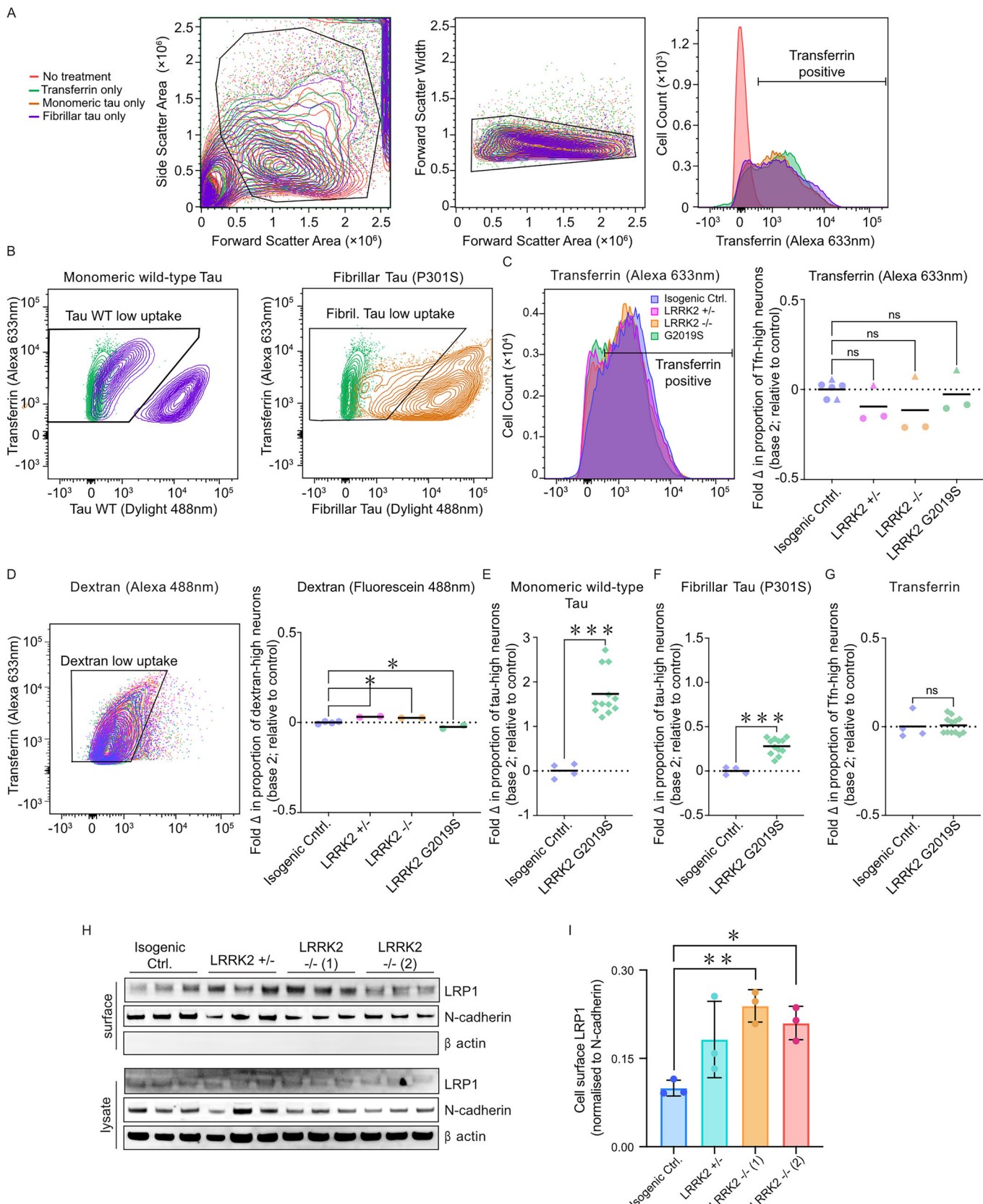

◀  **Figure EV6.  Uptake of transferrin and dextran by LRRK2 gain and loss of function mutation-expressing neurons.**

(A) Flow cytometry gating strategy for detection of transferrin and indicated cargo protein uptake into iPSC-derived isogenic control neurons (65 days after induction). Left panels show discrimination based on scatter parameters of neurons incubated with either no treatment (red) or extracellular transferrin (green), monomeric tau (orange) or fibrillar tau (blue). Black polygons and bold horizontal bars indicate gates applied to isolate the population of single neurons that endocytose transferrin. To establish threshold levels for positive protein uptake (bold horizontal bars), neurons without transferrin (red) or tau (green contours) incubation were analysed. (B) Fluorescent intensity of either monomeric or fibrillar Dylight 488 nm low tau protein uptake populations are indicated by black polygons. (C) Isogenic control, LRRK2 heterozygous ($+/-$), homozygous ($-/-$) null or LRRK2 G2019S mutant neurons (65 days after induction) were incubated with 90 nM transferrin (Tfn) Alexa 633 nm or Dextran Fluorescein 488 nm (D) for 4 h before dissociation into single cells and analysis by flow cytometry. For each genotype and cell line indicated, the percentage of cells with low protein uptake (indicated by the black outlined polygons) was compared to the mean (dashed line) of protein uptake of control neurons and displayed as fold change (log2). Significance was determined using one-way ANOVA ($*p < 0.05$, $**p < 0.01$, Dunnett's test for multiple comparisons; circles and triangles represent independent experiments; $n = >3$). (E–G) Biological replicate flow cytometry assays of the effect of LRRK2 G2019S mutation on neuronal uptake of extracellular tau. Isogenic control and LRRK2 G2019S mutant neurons (65 days after induction) were incubated with 90 nM transferrin (Tfn) Alexa 633 nm and either monomeric wild-type (E), or fibrillar P301S (F) tau Dylight 488 nm, or transferrin alone (G), for 4 h before dissociation into single cells and analysis by flow cytometry (Thermo Fisher Attune CytPix). For LRRK2 G2019S mutant neurons, the percentage of cells with low protein uptake (in neurons gated for transferrin uptake) was compared to the mean (dashed line) of protein uptake of control neurons and displayed as fold change (log2). Significance was determined using an unpaired *t*-test ($***p < 0.001$; $n = 4$ for controls and $n = 12$ for LRRK2 G2019S; neurons generated from two and four independent neural inductions from control and LRRK2 G2019S iPSCs, respectively). (H) LRRK2 homozygous null neurons have increased levels of tau receptor protein LRP1 at the neuronal surface compared with isogenic controls. LRRK2 heterozygous and homozygous null neurons (60 days after induction) were surface biotinylated, and Neutravidin-coated particles were used to capture biotinylated membrane proteins. Surface abundance of indicated proteins were measured by immunoblotting. (I) Cell surface levels of LRP1 normalised to N-cadherin are shown for each genotype. Significance was determined using one-way ANOVA ($*p < 0.05$, $**p < 0.01$, Dunnett's test for multiple comparisons, $n = 3$).

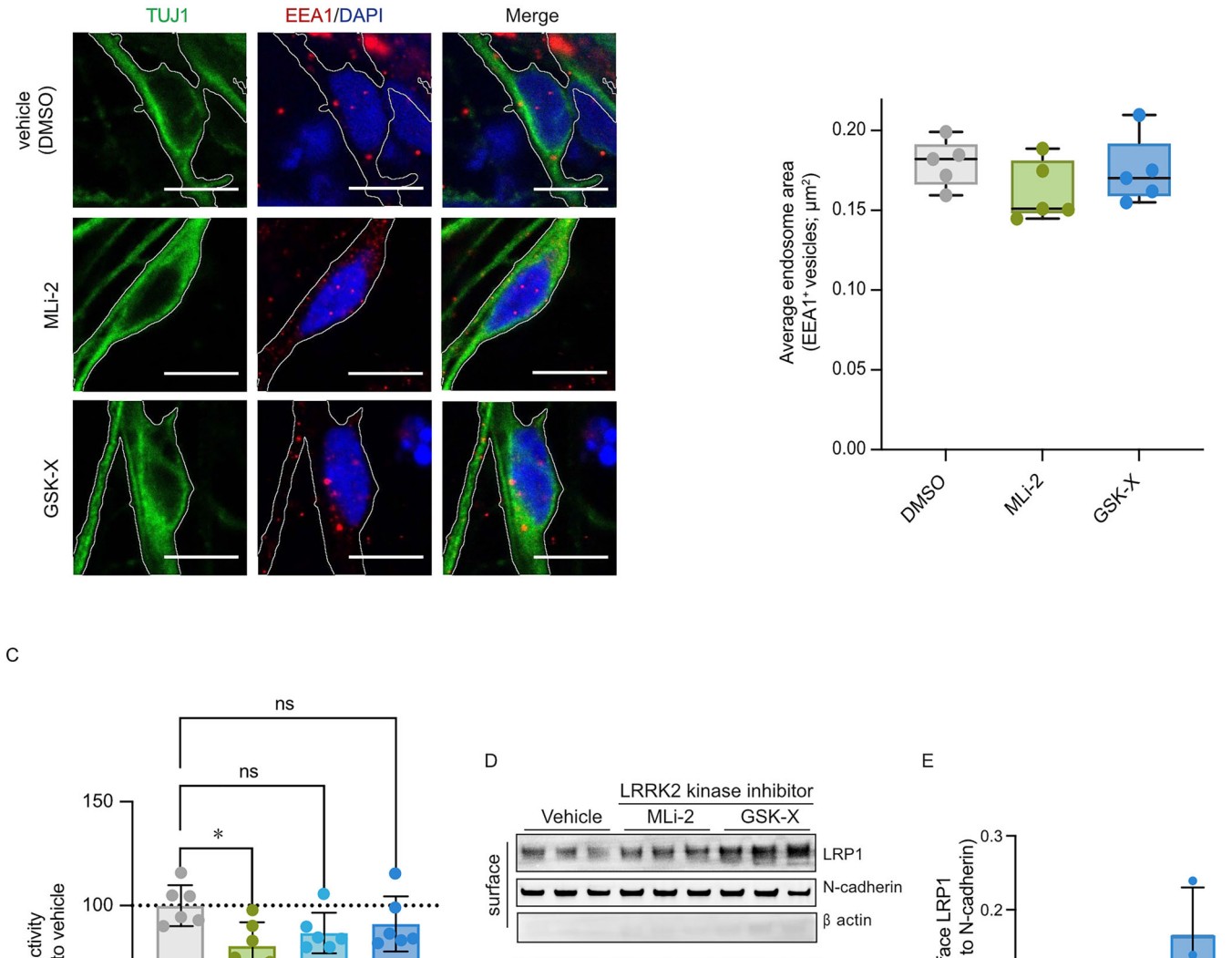

**Figure EV7. Effect of LRRK2 kinase activity inhibitors on the endolysosomal system of human cortical neurons.**

(A, B) Neurons treated with LRRK2 kinase activity inhibitors do not exhibit an endosomal or lysosomal phenotype. (A) Representative immunohistochemistry of neurons (60 days after induction) treated for 1 week with vehicle control (0.1% [v/v] DMSO), 1 μM MLi-2 or 10 μM GSK-X, immunostained for neuronal β3-tubulin (TUJ1, green), early endosomes (EEA1; red) and nuclei were counterstained (DAPI, blue). Scale bar, 10 μm. (B) No significant changes in the average size of early endosomes (EEA positive vesicles; μm²) in neurons compared with isogenic control ($n = >5$ images). (C) Extracellular LDH activity was used to assess neuronal viability in the presence of vehicles (0.1% [v/v] DMSO), 1 μM MLi-2, 1 μM or 10 μM GSK-X (GSK3357679A; treatment for 1 week). Only 1 μM MLi-2 had a modest effect on neuronal viability (six wells per treatment). Error bars indicate SD. Significance was determined using one-way ANOVA ($*p < 0.05$, Dunnett's test for multiple comparisons). (D) Effect of inhibition of LRRK2 kinase activity on neuronal surface levels of tau receptor protein LRP1. Neurons were pre-incubated with either 1 μM MLi-2, 10 μM GSK-X or vehicle control (0.1% [v/v] DMSO) for one week prior to cell surface biotinylation (61 days after induction), followed by capture of biotinylated membrane proteins using Neutravidin-coated particles. Surface abundance of indicated proteins were examined by immunoblotting. (E) Cell surface levels of LRP1 normalised to N-cadherin are shown for each of the treatments (three replicate treatments).

