## [Peer Review File · The EMBO Journal]

Monomeric tau uptake by human neurons depends on receptor LRP1 and kinase LRRK2

Lewis Evans, Alessio Strano, Eleanor Tuck, Ashley Campbell, James Smith, Christy Hung, Tiana Behr, Bernardino Ghetti, Benjamin Ryskeldi-Falcon, Emre Karakoc, Francesco Iorio, Alastair Reith, Andrew Bassett, and Frederick Livesey

Corresponding author(s): Frederick Livesey (rick@talisman-therapeutics.com)

Review Timeline:	Transfer Date:	27th Jan 25
	Editorial Decision:	3rd Mar 25
	Revision Received:	8th May 25
	Editorial Decision:	2nd Jun 25
	Revision Received:	2nd Jul 25
	Accepted:	3rd Jul 25

Editor: Ioannis Papaioannou

Transaction Report: This manuscript was transferred to

The EMBO Journal following peer review at Review Commons.

**Review
COMMONS**

Review #1

1. Evidence, reproducibility and clarity:

Evidence, reproducibility and clarity (Required)

The authors investigated the cellular uptake of tau in neurodegenerative diseases. Using a genome-wide CRISPR loss-of-function screening in human iPSC-derived excitatory neurons, they identified distinct cellular pathways involved in the uptake of extracellular monomeric and fibrillar tau. The screening results revealed that LRRK2, along with the previously recognized LRP1, plays a role in the uptake of monomeric tau. While LRP1 was critical for the uptake of monomeric tau, it did not contribute to the uptake of fibrillar tau. Similarly, the endocytosis of monomeric tau was dependent on the familial Parkinson's disease gene LRRK2, but LRRK2 was not required for the endocytosis of fibrillar tau. These findings suggest that LRP1 and LRRK2 are involved in the pathogenesis of tauopathies and Parkinson's disease, highlighting LRRK2 as a potential therapeutic target for these diseases.

1. Since all results rely on isogenic iPSC lines from only one donor, authors need to confirm their finding using iPSC lines from another donor.
2. There are no sufficient attempts to assess the effects on synaptic functions and neurotoxicity.
3. It is unclear how many technical replicates and how many independent experiments are performed in each experiment.
4. Since FACS may detect tau uptake in only soma, the effects of tau uptake should be evaluated by imaging entire neurons including axon and dendrites.
5. In addition to RAP and LRP1 domain 4, it should be considered validating the results using LRP1 KO models or knockdown approaches.
6. Detailed descriptions in the Methods section for the neuronal differentiation, reagent catalog numbers, reagent concentrations, experimental procedures, and analytical methods should be provided.
7. The concentrations and catalog numbers of RAP chaperone and LRP1 domain 4 is unclear
8. Individual data should be included as dots in all bar graphs.

2. Significance:

Significance (Required)

While their findings are interesting, there are several concerns which should be further addressed.

3. How much time do you estimate the authors will need to complete the suggested revisions:

Estimated time to Complete Revisions (Required)

(Decision Recommendation)

Between 3 and 6 months

Yes

Review #2

1. Evidence, reproducibility and clarity:

Evidence, reproducibility and clarity (Required)

This study describes genome-wide, FACS-based, pooled CRISPR knock-out screens carried out in human cortical neurons, to determine the cellular factors that are required for endocytosis of monomeric and fibrillar tau protein. The screens combined fluorescent tau species uptake with labelled transferrin endocytosis (which is predominantly clathrin-dependent). This allowed identification of genes that had specific effects on tau endocytosis versus general endocytosis.

The study identified a plethora of genes/proteins that are required for tau endocytosis. Bioinformatics analysis convincingly demonstrated that the genes required for uptake of both forms of tau are enriched for various endocytic machineries; there was a partial overlap, as well as some important differences, in the classes of machinery involved for monomeric versus fibrillar tau. Reassuringly, the screen for monomeric tau identified LPR1 as important for its endocytosis, consistent with the previous literature, and individual validation results for several other genes confirmed their effect.

Importantly, the study also identified LRRK2 as being important for the uptake of monomeric tau. Further experiments were carried out with gene edited neurons lacking

LRRK2, or expressing mutated LRRK2, to characterise this finding in more detail. These identified morphological abnormalities in the endolysosomal system, and also validated that LRRK2 regulates neuronal endocytosis of other key molecules that have been linked to neurodegenerative diseases, such as alpha-synuclein and Aβ. The precise mechanism of this effect of LRRK2 is not clear, and I'm sure will be a fruitful topic for additional studies; it is beyond the scope of the present study.

Overall, I think this is a well-conducted study that is nicely written with well-presented data. The data are largely convincing. The strengths of this study include that:

- the studies are carried out in human neurons, important target cells of tauopathies.
- the screen is nicely designed and the QC presented is thorough.
- it defines the landscape of cellular processes that are involved in tau endocytosis, a process that is highly likely to be of pathological relevance to major neurological disorders.
- an important mechanistic link between LRRK2 mutations and tau uptake is identified and further characterised.
- in virtually all cases (apart from a few experiments, e.g. Figure 6f), the studies are carried out with sufficient replicates and the statistical analysis is, as far as I can tell, appropriate (I do not have detailed experience in the statistical analysis of functional genomics datasets).

My criticisms of this study are all minor:

- in the initial QC of the screen, it would be interesting to see immunofluorescence microscopy assays with labelled tau species to further validate the FACS-based uptake assay is behaving as expected. At the time-point examined by FACS, is most tau in an endosomal compartment (as would be expected)? Furthermore, as an optional point for the authors to consider, in general I think the paper would be enhanced by inclusion of representative immunofluorescence images (as extended information) to supplement the FACS data in some of the subsequent figures, for example those in Figure 4a-d and Figure 6; although I think the conclusions of the paper are supported without such images, they would provide a nice visual representation of the effects.
- in Figure 2f there is validation of a selection of screen hits by targeted CRISPR knock-out of the genes involved and FACS-based assays. Was this done with different CRISPR guides to those used in the initial screen, to provide further reassurance that there are no off-target effects? In addition, depletion of the mRNA/protein of interest is not confirmed in these validation experiments and this should be shown.
- in figure 4, the LAMP1 labelling is poorly resolved and it is difficult to see how the surface

area of individual puncta could have been accurately measured. In addition, LAMP1 labelling is used as a proxy for the lysosomal compartment and I'm sure the authors appreciate that LAMP1 also labels late endosomal and autophagic compartments. I would suggest additional labelling for a lysosomal enzyme (e.g. cathepsin B or D) to provide additional specificity. This also tends to allow better delineation of individual vesicles than LAMP1, allowing easier measurement of lysosomal size.

- on page 12, regarding the vacuolar ATPase hits from the screen, referring to Figures 4c,d, it is stated that the results indicate "both forms of tau protein are trafficked via intracellular acid compartments of neurons". However, the function of the vacuolar ATPase has also been linked to effects on clathrin-mediated endocytosis (see PMID: 23263279) and this could provide a more direct explanation for the effect seen. This possibility should be mentioned. In addition, I think the authors overstate the case that the Brefeldin experiments "confirm" the dependency of tau uptake on ER-Golgi transport. Brefeldin was used for 24 hours and so there could be many knock-on effects of this treatment. The authors should either soften this statement or provide additional evidence (e.g. through other methods of blocking ER-Golgi and Golgi traffic such as depletion of individual key proteins involved in the process - which could be selected from the screen hits) to support it.

- in certain figures bar graphs are shown, and these would be improved if they also showed the individual replicate data points.

****Referees cross-commenting****

Re Reviewer 1's comments:

1. Since all results rely on isogenic iPSC lines from only one donor, authors need to confirm their finding using iPSC lines from another donor.

- Although the authors could consider this, I don't think this is strictly necessary. To my mind one of the key strengths of the study is that the lines used are isogenic, meaning that genetic background effects are controlled for. Perhaps the authors could deal with this by recognising this limitation of the study in the text.

2. There are no sufficient attempts to assess the effects on synaptic functions and neurotoxicity.

- I think that this is beyond the scope of the current study.

3. It is unclear how many technical replicates and how many independent experiments are performed in each experiment.

- This is a fair point. It can sometimes be a little moot as to what constitutes a replicate for a biological repeat in such cell biology experiments, and the authors should clarify more

clearly what they have done, and whether they consider it a replicate or biological repeat.

4. Since FACS may detect tau uptake in only soma, the effects of tau uptake should be evaluated by imaging entire neurons including axon and dendrites.

- I made a similar point in my review.

5. In addition to RAP and LRP1 domain 4, it should be considered validating the results using LRP1 KO models or knockdown approaches.

- The authors could consider this. My opinion was that two orthogonal approaches was sufficient.

6. Detailed descriptions in the Methods section for the neuronal differentiation, reagent catalog numbers, reagent concentrations, experimental procedures, and analytical methods should be provided.

- Agreed

7. The concentrations and catalog numbers of RAP chaperone and LRP1 domain 4 is unclear

- Agreed

8. Individual data should be included as dots in all bar graphs.

- Agreed

2. Significance:

Significance (Required)

In conclusion, I feel that this is an important study that provides a conceptual advance to the field, especially in delineating the landscape of cellular functions involved in tau endocytosis and in providing a mechanistic linkage between LRRK2 function and tau endocytosis, as well as the endocytosis of other key neurodegeneration-associated molecules. I think that it will be of interest to a broad readership, including basic and translational scientists in the fields of Alzheimer's and Parkinson diseases and other prevalent neurodegenerative disorders. I anticipate that this paper will provide information that stimulates many subsequent studies.

3. How much time do you estimate the authors will need to complete the suggested revisions:

Estimated time to Complete Revisions (Required)

(Decision Recommendation)

Between 1 and 3 months

4. Review Commons values the work of reviewers and encourages them to get credit for their work. Select 'Yes' below to register your reviewing activity at Web of Science

Reviewer Recognition Service (formerly Publons); note that the content of your review will not be visible on Web of Science.

Yes

Revision Plan

Manuscript number: RC-2024-02808

Corresponding author(s): Frederick Livesey

1. General Statements

We thank the reviewers for their positive assessment of our study, and their acknowledgment of the study's breadth and significance. We respond below to both reviewers' comments, incorporating the cross-commenting between the reviewers.

2. Description of the planned revisions

The combined reviews raise the following main points that we propose addressing in revision:

1. Validation that the FACS assay for tau uptake captures total uptake of tau (ie, from neurites as well as from the cell body)
2. Clarification of numbers of technical and biological replicates studied
3. Validation of LRP1's role in tau uptake by loss of function
4. Increased detail of methods and reagents used
Confirmation of successful targeting of individual genes in the replication studies

To address these points we propose the following revisions:

1. Validation that the FACS assay for tau uptake captures total uptake of tau (ie, from neurites as well as from the cell body)

This study builds upon our previous published work in which we established FACS and image-based approaches for measuring tau uptake (Evans et al., 2018). In this previous study we reported immunofluorescent images of internalised tau of the type suggested by the reviewer, as well as, live high-content imaging of tau uptake. This previous study found that pHrodo-labelled tau accumulated in neuronal cell bodies over several hours (eg, Figure 3 and associated movies in Evans et al., 2018). In this present study, we used similar experimental parameters to that previous work. We include representative images from our live high-content analysis in Extended Data Figure 2e that again show that pHrodo-labeled tau accumulates in neuronal cell bodies over the 4 hour imaging period. For this screen, we identified concentrations of monomeric and fibrillar tau, and time for uptake, in which we observed close to saturating levels of tau uptake by FACS (Extended Data Figure 1d,e). These conditions were chosen to maximise the number of neurons taking up tau in these somewhat dense culture systems, reducing the possibility of false-positive hits. Thus, the FACS-based assay, which measures labelled tau within the neuronal cell body, detects tau that has been internalised from both neurites and cell bodies. We have clarified this within the text.

Revision Plan

2. Clarification of numbers of technical and biological replicates studied

This has been carried out throughout the manuscript, clearly highlighting both technical and biological replicates.

3. Validation of LRP1's role in tau uptake by loss of function

This is included in Figure 2f, where LRP1 was knocked down by CRISPR in human neural progenitor cells. As reviewer 2 identifies, we additionally show a reduction in tau uptake in the presence of small molecules that specifically compete with LDL receptors. These data and the published study (Rauch et al., 2020) provide compelling evidence that LRP1 is involved in tau uptake. The successful targeting of LRP1 was confirmed with T7 endonuclease assay: those data are included in the revised manuscript.

4. Increased detail of methods and reagents used

This has now been carried out throughout the manuscript. We have modified methods to include a description of our neural differentiation protocol, including reagent catalogue numbers, concentrations and procedures.

5. Confirmation of successful targeting of individual genes in the replication studies

CRISPR targeting and disruption of six genes analysed in the replication studies reported in Figure 2 was confirmed using the T7 endonuclease assay (AP2M1, AP2S1, PIK3R4, CCDC115, LRP1 and COMMD10). Those data are now included in the Extended Data figure 2h.

3. Description of the revisions that have already been incorporated in the transferred manuscript

All of the revisions listed above have been added to the transferred manuscript, and also modified the text as suggested by Reviewer 2 to clarify a small number of points.

4. Description of analyses that authors prefer not to carry out

Replicating key findings in additional iPSC lines

Reviewer 1 suggests replicating the main findings of the study in additional iPSC lines. However, Reviewer 2 does not consider necessary, highlighting the use of isogenic iPSC lines as a strength of the study. We concur with Reviewer 2 using the same genetic background, limiting variance arising from differences in differentiation potential among different starting iPSCs, a common issue. Additionally, we used multiple iPSC clones for the validation of our LRRK2 findings, accounting for non-specific off-target effects of gene editing.

Revision Plan

Analysis of lysosome function in LRRK2 mutant neurons with fluorescent reporters (Figure 4)

In parallel studies, we have performed imaging assays in human iPSC-derived neurons using Pepstatin A-BODIPY FL and Magic Red Cathepsin B. In these relative dense cultures, while these reagents visualise punctae somewhat better than LAMP1 immunocytochemistry, we did not observe a substantial improvement: in both cases individual lysosomes were challenging to resolve with confidence. Therefore, we have not included such experiments here, and we have now clarified in the text that we are measuring total LAMP1-positive material/mass, which includes late endosomes and some autophagosomes, rather than individual lysosomes in those experiments.

Tau protein neurotoxicity

Reviewer 1 suggests assessment of neurotoxicity, whereas Reviewer 2 does not consider this necessary. Our previous study (Evans et al., *Cell Reports*, 2018) included an assessment of tau protein neurotoxicity in our system. Over the course of the experiment (<5hrs) we did not observe neurotoxicity. In this new study, we include lactate dehydrogenase (LDH) assay measurements to assess cell viability in the presence of different forms of tau protein, including postmortem tau (Extended Data Figure 2f) and the small molecule treatments in this study (Extended data figures 2g, 4a,b). In parallel work, we have performed experiments where tau protein has been applied to culture for approximately a week, with no detrimental effect to neuronal survival.

Dear Rick,

Thank you again for transferring your revised manuscript (EMBOJ-2025-120302-T) from Review Commons to The EMBO Journal, along with the reports of the referees who had assessed its previous version there. Your revision has been sent back to the same reviewers for a second round of review, and we have now received their comments, which I have already shared with you (they are included again below).

I am very pleased to say that they are both very satisfied with the revision, recognize the conceptual advance and solidity of the study, and are now supportive of publication of the manuscript in The EMBO Journal.

From the editorial side, there are a few corrections and changes that we need you to make in a final version of the manuscript before we can proceed with its formal acceptance for publication. Please include in your resubmission a cover letter detailing how the points below are addressed and all changes to the manuscript:

- Please note that the funding information should be provided in the Acknowledgements section of the manuscript file, not in a separate "Funding" section. In addition, the same funding information should be provided both in our online manuscript tracking system (eJP) and in the Acknowledgements section of the revised manuscript; currently, there are some inconsistencies between the two, namely missing information in the manuscript file: grant number WT101052MA for Wellcome Trust; and missing information in eJP: Dementias Platform UK (Stem Cell Network) and Great Ormond Street Hospital Charity.
- Please provide a list of up to 5 relevant keywords after the Abstract of your revised manuscript.
- The reference format is not correct. Our format is alphabetical, and "et al." follows the names of the first 10 co-authors in cases of publications with more than 10 co-authors. You can find more information about our citation format in our guide to authors: <https://www.embopress.org/page/journal/14602075/authorguide#referencesformat>.
- Before submitting your revision, primary datasets (and computer code, where appropriate) produced in this study need to be deposited in appropriate public databases (see <https://www.embopress.org/page/journal/14602075/authorguide#dataavailability>). Their accession numbers, databases, and the specific URLs (links) should be listed in a formal "Data availability" section (placed after Methods).

In particular, we kindly request that the RNA-sequencing data produced in the study be deposited in a public repository, and that access information be provided in the Data availability section.

*** All links should resolve to a page where the data can be accessed. ***

*** The Data availability statement is restricted to new primary data that are part of the study. Please use data citations in the References list for already available datasets that were re-analyzed in your study instead of describing them in the Data availability section. ***

- Please change the heading "Competing interests" to "Disclosure and competing interests statement".
- As per our journal's policy, "data not shown" (on page 21) is not permitted. All data referred to in the paper should be displayed in the main or Expanded View figures, or in the Appendix. Please add these data or change the text accordingly if these data are not central to the study and its conclusions, or properly cite the respective published sources if these data can be found elsewhere.
- We noticed that figure callouts are missing for Figures 2A and 3C. Please make sure that all Figure panels are called out in your revised manuscript.
- When you are ready to submit your revision, please also upload a complete author checklist, which you can download from our author guidelines (<https://www.embopress.org/page/journal/14602075/authorguide>). Please note that the checklist will also be part of the Peer Review File.
- The figures should be uploaded as individual high-resolution Figure files with their legends included in the manuscript file (after the References list). All source file names, titles, legends, and manuscript callouts for supplementary figures need to be updated to "Figure EV1-EV7".
- All source file names, titles, legends, and manuscript callouts for Supplementary Table 2, 3, 5 and 6 all need to be updated to "Dataset EV1-EV4" with their legends removed from the manuscript file and provided instead in an individual tab/sheet in each corresponding Excel file.
- All source file names, titles, legends, and manuscript callouts for Supplementary Table 1 and 4 need to be updated to "Table

EV1-EV2" with their legends removed from the manuscript file and provided instead above each table.

- The materials and methods need to be described in the manuscript using our structured methods format, which is now required for all research articles. According to this format, the Methods section includes a single "Reagents and Tools Table" -listing key reagents, experimental models, software and relevant equipment including their sources and relevant identifiers- followed by a "Methods and Protocols" section describing the methods. Please download and fill our Reagents and Tools Table template (.docx), which you can find in our author guide:

<https://www.embojournal.org/page/journal/14602075/authorguide#structuredmethods>. When submitting your revised manuscript, please do not include the Reagents and Tools Table in the Methods section of the manuscript but instead upload it as a separate file choosing the file type "Reagent Table".

- At EMBO Press we ask authors to provide source data for the main manuscript Figures. Our source data coordinator has already contacted you separately discussing which Figure panels we would need source data for and also providing you with helpful tips on how to upload and organize the files.

- Please note that EMBO press papers are accompanied online by:

A) a short (2 sentences) summary of the findings and their significance,

B) 2-5 short bullet points highlighting the key results, and

C) a synopsis image in .jpg or .png format that is exactly 550 pixels wide and 300-600 pixels high (the height is variable). Please note that the text needs to be legible at the final size.

Please upload this information along with your revised manuscript (the text for A and B should be provided in a separate Word file).

- During our routine pre-acceptance checks, our data editors have raised the following queries regarding figures, data, and legends. Please make sure that all requests below are completely addressed in the final version of your manuscript:

1. Please provide the exact p values in the legends of Figures 2F, 4C, D, E; 6A-F; 7C, D.

2. Please indicate the statistical test used for data analysis in the legends of Figures 2F, 3B.

3. Please indicate what */ **/ ***/ **** represents; if this represents p value(s), please indicate the statistical test used and where appropriate, and the exact p value in the legend(s) of Figure(s) 2G, 4A, B; 5B, D, F, H.

4. Please note that the box plots need to be defined in terms of minima, maxima, centre, bounds of box and whiskers, and percentile in the legends of figures 5B, D, F, H; 7B.

- The manuscript section order should be corrected as follows: Title page - Abstract & Keywords - Introduction - Results - Discussion - Methods - Data Availability - Acknowledgements - Disclosure and Competing Interests Statement - References - Figure Legends - main Table(s) (if there are any) - Expanded View Figure Legends.

Please also note that as part of the EMBO publications' Transparent Editorial Process, The EMBO Journal publishes online a Peer Review File along with each accepted manuscript. This File will be published in conjunction with your paper and will include the referee reports, your point-by-point response and all pertinent correspondence relating to the manuscript. You can opt out of this by letting the editorial office know (contact@embojournal.org). If you do opt out, the Peer Review File link will point to the following statement: "No Peer Review File is available with this article, as the authors have chosen not to make the review process public in this case."

We look forward to seeing a final version of your manuscript as soon as possible. Please let us know if you have any questions and use this link to submit your revision: Unavailable.

Best regards,

Ioannis

Referee #1:

The authors have sufficiently addressed the reviewers' concerns.

Referee #2:

General Summary and Significance:

This study describes genome-wide, FACS-based, pooled CRISPR knock-out screens carried out in human cortical neurons, to determine the cellular factors that are required for endocytosis of monomeric and fibrillar tau protein. The screens combined fluorescent tau species uptake with labelled transferrin endocytosis (which is predominantly clathrin-dependent). This allowed identification of genes that had specific effects on tau endocytosis versus general endocytosis.

The study identified a plethora of genes/proteins that are required for tau endocytosis. Bioinformatics analysis convincingly demonstrated that the genes required for uptake of both forms of tau are enriched for various endocytic machineries; there was a partial overlap, as well as some important differences, in the classes of machinery involved for monomeric versus fibrillar tau. Reassuringly, the screen for monomeric tau identified LPR1 as important for its endocytosis, consistent with the previous literature, and individual validation results for several other genes confirmed their effect.

Importantly, the study also identified LRRK2 as being important for the uptake of monomeric tau. Further experiments were carried out with gene edited neurons lacking LRRK2, or expressing mutated LRRK2, to characterise this finding in more detail. These identified morphological abnormalities in the endolysosomal system, and also validated that LRRK2 regulates neuronal endocytosis of other key molecules that have been linked to neurodegenerative diseases, such as alpha-synuclein and Abeta. The precise mechanism of this effect of LRRK2 is not clear, and I'm sure will be a fruitful topic for additional studies; it is beyond the scope of the present study.

Overall, I think this is a well-conducted study that is nicely written with well-presented data. The data are largely convincing. The strengths of this study include that:

- the studies are carried out in human neurons, important target cells of tauopathies.
- the screen is nicely designed and the QC presented is thorough.
- it defines the landscape of cellular processes that are involved in tau endocytosis, a process that is highly likely to be of pathological relevance to major neurological disorders.
- an important mechanistic link between LRRK2 mutations and tau uptake is identified and further characterised.
- in virtually all cases (apart from a few experiments, e.g. Figure 6f), the studies are carried out with sufficient replicates to enable statistical analysis. A caveat is that in many experiments this relies on technical repeats rather than biological repeats, but in totality the experiments are convincing.

In conclusion, I feel that this is an important study that provides a conceptual advance to the field, especially in delineating the landscape of cellular functions involved in tau endocytosis and in providing a mechanistic linkage between LRRK2 function and tau endocytosis, as well as the endocytosis of other key neurodegeneration-associated molecules. I think that it will be of interest to a broad readership, including basic and translational scientists in the fields of Alzheimer's and Parkinson diseases and other prevalent neurodegenerative disorders. I anticipate that this paper will provide information that stimulates many subsequent studies.

Concerns:

The authors have dealt with my previous criticisms of the paper (via Review Commons) in a satisfactory way, so I have no further criticisms.

Rev_Com_number: RC-2024-02808

New_manu_number: EMBOJ-2025-120302-T

Corr_author: Livesey

Title: Whole genome CRISPR screens identify a LRRK2-regulated pathway for tau uptake by human neurons

CONFIRMATION OF FINAL CORRECTIONS

- Please note that the funding information should be provided in the Acknowledgements section of the manuscript file, not in a separate "Funding" section. In addition, the same funding information should be provided both in our online manuscript tracking system (eJP) and in the Acknowledgements section of the revised manuscript; currently, there are some inconsistencies between the two, namely missing information in the manuscript file: grant number WT101052MA for Wellcome Trust; and missing information in eJP: Dementias Platform UK (Stem Cell Network) and Great Ormond Street Hospital Charity.

Now included.

- Please provide a list of up to 5 relevant keywords after the Abstract of your revised manuscript.

Now included.

- The reference format is not correct. Our format is alphabetical, and "et al." follows the names of the first 10 co-authors in cases of publications with more than 10 co-authors. You can find more information about our citation format in our guide to authors:

<https://www.embopress.org/page/journal/14602075/authorguide#referencesformat>

References now conform to EMBO Press format.

- Before submitting your revision, primary datasets (and computer code, where appropriate) produced in this study need to be deposited in appropriate public databases (see

<https://www.embopress.org/page/journal/14602075/authorguide#dataavailability>). Their accession numbers, databases, and the specific URLs (links) should be listed in a formal "Data availability" section (placed after Methods).

In particular, we kindly request that the RNA-sequencing data produced in the study be deposited in a public repository, and that access information be provided in the Data availability section.

*** All links should resolve to a page where the data can be accessed. ***

*** The Data availability statement is restricted to new primary data that are part of the study. Please use data citations in the References list for already available datasets that were re-analyzed in your study instead of describing them in the Data availability section. ***

Not applicable for sequencing – no relevant data, including RNA sequencing data, were generated for this study. Source data now uploaded to BioStudies, accession number S-BSST2010.

- Please change the heading "Competing interests" to "Disclosure and competing

interests statement".

Now corrected.

- As per our journal's policy, "data not shown" (on page 21) is not permitted. All data referred to in the paper should be displayed in the main or Expanded View figures, or in the Appendix. Please add these data or change the text accordingly if these data are not central to the study and its conclusions, or properly cite the respective published sources if these data can be found elsewhere.

Now corrected.

- We noticed that figure callouts are missing for Figures 2A and 3C. Please make sure that all Figure panels are called out in your revised manuscript.

Now corrected.

- When you are ready to submit your revision, please also upload a complete author checklist, which you can download from our author guidelines (<https://www.embopress.org/page/journal/14602075/authorguide>). Please note that the checklist will also be part of the Peer Review File.

Included in submission.

- The figures should be uploaded as individual high-resolution Figure files with their legends included in the manuscript file (after the References list). All source file names, titles, legends, and manuscript callouts for supplementary figures need to be updated to "Figure EV1-EV7".

Completed.

- All source file names, titles, legends, and manuscript callouts for Supplementary Table 2, 3, 5 and 6 all need to be updated to "Dataset EV1-EV4" with their legends removed from the manuscript file and provided instead in an individual tab/sheet in each corresponding Excel file.

Now corrected.

- All source file names, titles, legends, and manuscript callouts for Supplementary Table 1 and 4 need to be updated to "Table EV1-EV2" with their legends removed from the manuscript file and provided instead above each table.

Now corrected.

- The materials and methods need to be described in the manuscript using our structured methods format, which is now required for all research articles. According to this format, the Methods section includes a single "Reagents and Tools Table" -listing key reagents, experimental models, software and relevant equipment including their sources and relevant identifiers- followed by a "Methods and Protocols" section describing the methods. Please download and fill our Reagents and Tools Table template (.docx), which you can find in our author guide:

<https://www.embopress.org/page/journal/14602075/authorguide#structured>

methods. When submitting your revised manuscript, please do not include the Reagents and Tools Table in the Methods section of the manuscript but instead upload it as a separate file choosing the file type "Reagent Table".

Now corrected.

- At EMBO Press we ask authors to provide source data for the main manuscript Figures. Our source data coordinator has already contacted you separately discussing which Figure panels we would need source data for and also providing you with helpful tips on how to upload and organize the files.

Included as part of resubmission.

- Please note that EMBO press papers are accompanied online by:

- A) a short (2 sentences) summary of the findings and their significance,
- B) 2-5 short bullet points highlighting the key results, and
- C) a synopsis image in .jpg or .png format that is exactly 550 pixels wide and 300-600 pixels high (the height is variable). Please note that the text needs to be legible at the final size.

Please upload this information along with your revised manuscript (the text for A and B should be provided in a separate Word file).

Included as part of resubmission.

- During our routine pre-acceptance checks, our data editors have raised the following queries regarding figures, data, and legends. Please make sure that all requests below are completely addressed in the final version of your manuscript:

1. Please provide the exact p values in the legends of Figures 2F, 4C, D, E; 6A-F; 7C, D.
2. Please indicate the statistical test used for data analysis in the legends of Figures 2F, 3B.
3. Please indicate what */ **/ ***/ **** represents; if this represents p value(s), please indicate the statistical test used and where appropriate, and the exact p value in the legend(s) of Figure(s) 2G, 4A, B; 5B, D, F, H.
4. Please note that the box plots need to be defined in terms of minima, maxima, centre, bounds of box and whiskers, and percentile in the legends of figures 5B, D, F, H; 7B.

Now corrected.

- The manuscript section order should be corrected as follows: Title page - Abstract & Keywords - Introduction - Results - Discussion - Methods - Data Availability - Acknowledgements - Disclosure and Competing Interests Statement - References - Figure Legends - main Table(s) (if there are any) - Expanded View Figure Legends.

Now corrected.

Please also note that as part of the EMBO publications' Transparent Editorial Process, The EMBO Journal publishes online a Peer Review File along with each accepted manuscript. This File will be published in conjunction with your paper

and will include the referee reports, your point-by-point response and all pertinent correspondence relating to the manuscript. You can opt out of this by letting the editorial office know (contact@embojournal.org). If you do opt out, the Peer Review File link will point to the following statement: "No Peer Review File is available with this article, as the authors have chosen not to make the review process public in this case."

Noted.

Dear Rick,

Thank you for addressing the majority of our editorial and formatting requests and for submitting your revised manuscript (EMBOJ-2025-120302R) to The EMBO Journal. We have now checked the latest version of your manuscript and are glad to confirm that all previous issues have been resolved, with only few exceptions that we kindly request you to also address as soon as possible so that we can formally accept your manuscript for publication in The EMBO Journal without any further delays.

In particular, we noticed that the Illumina sequencing data produced in the study have not been deposited to a public repository, or the access information is not provided in the Data availability statement. Please deposit all data to an appropriate database and provide the identity of the database, the dataset ID, and the specific and permanent URL in your Data availability section, according to the instructions that I have previously shared with you.

In addition, we note that we do not have access to your Source Data that have been deposited to BioStudies, as we were unable to access/download the data using the identifier provided in your Data availability statement. Please make sure that all data are publicly available, and also provide the complete URL of the dataset in your revised Data availability section.

Finally, we previously shared with you a list of requests regarding Figures/legends and data/statistics. We noticed that not all of them have been addressed, and we note that we cannot move forward with acceptance of the manuscript before they have all been fully addressed. For your convenience, they are included here again:

- During our routine pre-acceptance checks, our data editors have raised the following queries regarding figures, data, and legends. Please make sure that all requests below are completely addressed in the final version of your manuscript:
- 1. Please provide the exact p values in the legends of Figures 2F, 4C, D, E; 6A-F; 7C, D.
- 2. Please indicate the statistical test used for data analysis in the legends of Figures 2F, 3B.
- 3. Please indicate what */ **/ ***/ **** represents; if this represents p value(s), please indicate the statistical test used and where appropriate, and the exact p value in the legend(s) of Figure(s) 2G, 4A, B; 5B, D, F, H.
- 4. Please note that the box plots need to be defined in terms of minima, maxima, centre, bounds of box and whiskers, and percentile in the legends of figures 5B, D, F, H; 7B.

Please address these remaining issues in a final version of your manuscript, and upload it to our manuscript tracking system as soon as possible. We will be happy to formally accept your paper and proceed with its publication as soon as these issues have been fully resolved.

Best regards,

Ioannis

Rev_Com_number: RC-2024-02808

New_manu_number: EMBOJ-2025-120302R

Corr_author: Livesey

Title: Whole genome CRISPR screens identify a LRRK2-regulated pathway for tau uptake by human neurons

All editorial and formatting issues were resolved by the authors.

Dear Rick,

Congratulations on an excellent study! I am very pleased to inform you that your manuscript has been accepted for publication in The EMBO Journal. Thank you for comprehensively addressing the initially raised referees' concerns and all editorial requests for changes and corrections.

If you have any questions, please do not hesitate to contact the Editorial Office. Thank you for your contribution to The EMBO Journal. Working with you has been a pleasure!

Best regards,

Ioannis

Rev_Com_number: RC-2024-02808

New_manu_number: EMBOJ-2025-120302R1

Corr_author: Livesey

Title: Whole genome CRISPR screens identify a LRRK2-regulated pathway for tau uptake by human neurons